# QUANTIFYING TASK COMPLEXITY THROUGH GENERALIZED INFORMATION MEASURES

## ABSTRACT

How can we measure the "complexity" of a learning task so that we can compare one task to another? From classical information theory, we know that entropy is a useful measure of the complexity of a random variable and provides a lower bound on the minimum expected number of bits needed for transmitting its state. In this paper, we propose to measure the complexity of a learning task by the minimum expected number of questions that need to be answered to solve the task. For example, the minimum expected number of patches that need to be observed to classify FashionMNIST images. We prove several properties of the proposed complexity measure, including connections with classical entropy and sub-additivity for multiple tasks. As the computation of the minimum expected number of questions is generally intractable, we propose a greedy procedure called "information pursuit" (IP), which selects one question at a time depending on previous questions and their answers. This requires learning a probabilistic generative model relating data and questions to the task, for which we employ variational autoencoders and normalizing flows. We illustrate the usefulness of the proposed measure on various binary image classification tasks using image patches as the query set. Our results indicate that the complexity of a classification task increases as signal-to-noise ratio decreases, and that classification of the KMNIST dataset is more complex than classification of the FashionMNIST dataset. As a byproduct of choosing patches as queries, our approach also provides a principled way of determining which pixels in an image are most informative for a task.

## 1 INTRODUCTION

Deep networks have shown remarkable progress in both simple and complex machine learning tasks. But how does one measure the "complexity" of a learning task? Is it possible to ascertain in a principled manner which tasks are "harder" to solve than others? How "close" is one task to another? Answers to these questions would have implications in many fields of machine learning such as transfer learning, multi-task learning, un/semi/self-supervised learning, and domain adaptation.

In classical information theory, the entropy of a random variable $X$ is a useful measure of complexity for tasks such as compression and transmission, which essentially require reconstructing $X$. However, the entropy of $X$ is insufficient for measuring the complexity of a supervised learning task $T_{X,Y}$, where the goal is to predict an output $Y$ from an input $X$, i.e., to estimate the conditional $p_{Y|X}(y \mid x)$ from a finite set of samples from $p_{XY}(x, y)$, which we refer to as solving the learning task. Complexity measures provided by statistical learning theory like VC-dimension or covering numbers are also inadequate for this purpose because they ignore the dependence between $X$ and $Y$ for the particular task at hand. Information-theoretic measures such as *mutual information*, *information bottleneck* (Tishby et al., 2000) and its variants (Strouse & Schwab, 2017) have been used to study the trade-off between model complexity and accuracy, but have not been developed to focus on assessing task complexity and can provide unsatisfactory results when comparing different tasks (see Section 5 for details). Measures based on Kolmogorov complexity (Li, 2006; Vereshchagin & Vitányi, 2004) could in principle be used to compare different tasks, but they are dataset permutation sensitive and not easily computable. The work of (Achille et al., 2019a) proposes to quantify task complexity by measuring the information stored on the network weights, but the approach depends on the specific neural network architecture used for training. The work of (Tran et al., 2019) does not require or assume trained models, but makes strict assumptions that limit its broad applicability.

In this work, we introduce a novel perspective on task complexity which generalizes classical measures from information theory. Specifically, one well-known interpretation of classical Shannon entropy is, given a random variable $X$, find the minimum number of bits that are needed on average to encode instances of $X$ so that the instances can be perfectly recovered from the binary code. Stated differently, if one lets $Q$ be defined as the set of all possible binary functions, on the domain of $X$, then Shannon entropy essentially asks what is the optimal sequence of queries to compute $\{q_1(X), q_2(X), \ldots : q_i \in Q\}$ (i.e., how to encode $X$ as a binary string) so that $X$ can be perfectly recovered from the shortest (on average) sequence of binary answers to the queries (see Section 2 for more discussion of this interpretation). As discussed above, however, in most learning tasks we are not interested in simply compressing $X$ but rather making a prediction about some other variable $Y$. Further, notions of complexity can potentially be made more relevant to a specific task by not having $Q$ to be the set of all possible binary functions on $X$ but rather a smaller set of queries specific to a measure of interest. From this intuition, we define the complexity of a learning task as the minimum expected number of queries, selected from a fixed set $Q$, one needs to ask to predict $Y$ (to some user-defined level of confidence) from the respective answers to the queries. As a few specific examples of potential query sets:

- **Decision boundary complexity:** Here, $Q$ is the the set of all possible half-spaces in $\mathbb{R}^d$ (assuming $X \in \mathbb{R}^d$) and $q(x)$ is a binary function response indicating whether $x$ lies in a particular half-space ($q \in Q$). Then task complexity formalizes the intuition of "level of non-linearity" of the decision boundary. For example, the complexity of any linearly-separable binary classification task is 1, whereas, for a non-linearly separable task, this value depends on the curvature of the decision boundary.

- **Task-specific input feature complexity:** Here, $Q$ is the set of projection functions of $X$ and $q$ is of the form $q(x) = x_q$, where $x_q$ is the value observed at the $q^{th}$ entry of $x$. Then task complexity formalizes the intuition "the greater the redundancy between the input features the easier it is to solve the task". For example, $Y$ being a constant function of $X$ would be the simplest task with complexity 0, since no feature needs to be queried to predict it. This notion of complexity would help in answering questions such as "which input features are most important for solving a given task?" and could in turn help in developing more "interpretable" learning algorithms.

- **Visual semantic complexity:** Given a vocabulary $V$ of different possible entities, their attributes and relations in a visual scene, $Q$ could be the set of all binary functions indicating the presence or absence of an entity, its attribute or its relation with other entities (supplied by $V$) in a designated region of the image $X$. For example, a particular $q$ could be the function implementing the query "Is there a person in the top left corner of the image?". This notion of complexity would allow one to gauge the semantic complexity of a visual task. For instance, tasks which ask complex questions like "Is there a person playing with his dog, next to a river in the image?" would inherently be more complex than simple object detection tasks, "Where is the dog in this image?" and could be quantified by semantically relevant queries.

While our proposed formal definition of task complexity will be applicable to all such choices of query functions $\{q\}_{q \in Q}$ and enjoys several nice theoretical properties that we discuss in section 2, its computation will generally be intractable. As a result, in section 3 we propose to reduce the complexity of selecting a minimal set of questions by using the Information Pursuit (IP) algorithm, which selects questions sequentially, depending on previous questions and answers, in order of information gain. While IP is generally applicable to any task and query set, its implementation is still intractable depending on the complexity of the model $p(X, Y)$ and of the set $Q$. To address this issue, we posit a tractable graphical model for $p(X, Y)$ and learn the required distributions using variational autoencoders and normalizing flows. In section 4 we evaluate our approach on various binary image classification tasks (MNIST, KMNIST, FashionMNIST, Caltech Silhouettes) that can be tackled using a common set of queries (the set of image patches). Our results show that complexity computed using patch queries aligns with the intuition that the complexity of a classification task increases as signal-to-noise ratio decreases, and that classification of the KMNIST dataset is more complex than classification of the FashionMNIST dataset, something that isn't obvious a priori. While these experiments are restricted to simple tasks and queries, the proposed framework is generally applicable provided that tractable models, inference and learning methods can be developed, which is the subject of ongoing and future work. Finally, we note that to the best of our knowledge, this is the first time that a subjective notion of task complexity has been proposed in literature, where the user can incorporate in $Q$ the perception of complexity he/she wishes to measure.

## 2 QUANTIFYING TASK COMPLEXITY

Let the input data be represented by random variable $X$ and the corresponding output/hypothesis by random variable $Y$ with sample spaces $\mathcal{X}$ and $\mathcal{Y}$ respectively. A supervised learning task $T_{X;Y}$ is defined as the task of estimating the conditional distribution $p_{Y|X}(y \mid x)$ from a finite set of samples from the joint distribution $p_{XY}(x, y)$.[1] We propose to quantify the complexity of a task as the minimum expected number of queries needed to solve the task. Queries, $q \in Q$, are functions of the input data, whose answers for a given $x$ are denoted as $\{q(x)\}_{q \in Q}$. Naturally, the query set needs to be sufficiently rich so that the task is solvable based on answers to the queries. More formally, we say the set $Q$ is sufficient for task $T_{X;Y}$ if $\forall (x, y) \in \mathcal{X} \times \mathcal{Y}$,

$$p(y \mid x) = p(y \mid \{x' \in \mathcal{X} : q(x') = q(x) \, \forall q \in Q\}). \tag{1}$$

In other words, $Q$ is sufficient for task $T_{X;Y}$ if whenever two points $x$ and $x'$ have identical answers for all queries in $Q$, then their true posteriors must be equal, $p(y \mid x) = p(y \mid x')$.

Given a fixed query set $Q$, we now formally define an encoding function $E_Q$, which we refer to as a Q-Coder, along with our proposed complexity measure $C_Q(X, Y)$ for task $T_{X,Y}$.

**Defining the Encoder function.** Given a query set $Q$, an Encoder is a function, $E : S^* \to Q$ where $S^*$ is the set of all finite-length sequences generated using elements from the set $S = \{(q, q(x)) \mid q \in Q, x \in \mathcal{X}\}$. Additionally, we require that $Q$ contains a special query $q_{STOP}$ which signals the encoder to stop asking queries and outputs the code for $x$ as the set of query-answer pairs that have been asked. The process can be described as follows, given $E$ and input sample $x$.

1. $q_1 = E(\emptyset)$. The first question is independent of $x$.
2. $q_{k+1} = E(\{q_i, q_i(x)\}_{1:k})$. All subsequent queries depend on the query-answer pairs observed so far for $x$.
3. If $q_{L+1} = q_{STOP}$ terminate and return $Code_Q^E(x) := (q_i, q_i(x))_{1:L}$ as the code for $x$.

Note that each $q_i$ depends on $x$, but we drop this in the notation for brevity. Note also that the number of queries $L = |Code_Q^E(x)|$ for a particular $x$ generalizes *coding length* in information theory. The query $q_{STOP}$ constrains the code $Code_Q^E(x)$ to be prefix-free.

**Defining task complexity.** Given a joint distribution $p_{XY}(x, y)$ and a sufficient query set $Q$, we define task complexity, $C_Q(X; Y)$, as the minimum over all encoders $E$ (which are mappings from $X$ to a subset of queries in $Q$) of the mean coding length:

$$C_Q(X; Y) := \min_E \mathbb{E}_X \left[ |Code_Q^E(X)| \right]$$

$$\text{s.t.} \quad p(y \mid x) = p(y \mid Code_Q^E(x)) \; \forall x \in \mathcal{X}, y \in \mathcal{Y} \quad \textit{(Sufficiency)} \tag{2}$$

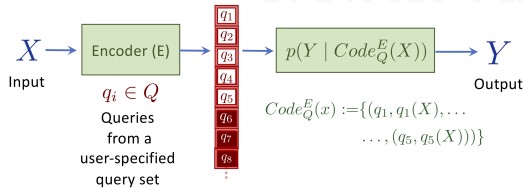

The first constraint ensures sufficiency of the code $\forall x \in \mathcal{X}$. By this we mean that there exists at least one Encoder $E$ for which the first constraint of "sufficiency" is satisfied, where $p(y \mid Code_Q^E(x))$ should be interpreted as the conditional probability of $y$ given the event $\{x' \in \mathcal{X} \mid Code_Q^E(x) = Code_Q^E(x')\}$. The solution to (2) provides the optimal encoder for task $T_{X;Y}$, and Fig. 1 illustrates the overall framework in detail.

Figure 1: *Schematic view of the overall framework for quantifying complexity of a task $T_{X;Y}$.*

**Connection with Shannon entropy $H$.** It can be shown that when $Q$ is taken as the set for all possible binary questions on $X$ and if $Y$ is a function of $X$ (which it usually is for supervised classification problems (Kolchinsky et al., 2018)), then the solution, $E^*$, to (2) gives a coding length within one bit of the entropy of $Y$, denoted as $H(Y)$. More formally, one can show that

$$H(Y) \leq C_Q(X; Y) = \mathbb{E}_X \left[ |Code_Q^{E^*}(X)| \right] < H(Y) + 1. \tag{3}$$

---

[1] As commonly used, we denote random variables by capital letters and their realizations with small letters.

Note that one example of such an optimal encoder, $E^*(x)$, is given by the Huffman code for $Y$.

**Connection with task complexity & equivalence classes.** Notions of complexity of an entity/object, often relate to the level of structural regularity present in the entity (Gell-Mann, 2002). $C_Q(X;Y)$ measures the degree of regularity present in the structure of $T_{X;Y}$. This structure is implicitly defined by the conditional $p_{Y|X}(y \mid x)$ and the query set $Q$. Notice that any $E$ partitions $\mathcal{X}$ into equivalence classes.

$$[x] = \{x' \in \mathcal{X} \mid Code_Q^E(x) = Code_Q^E(x')\}. \tag{4}$$

The prefix-free constraint on the codes generated by $E$ (due to ) ensures that $\forall x' \in [x], \forall y \in \mathcal{Y}, p_{Y|X=x'}(y) = p_{Y|X=x}(y)$. It is then natural to relate task complexity with the number of equivalence classes induced by the optimal $E^*$. The greater the number of equivalence classes, the higher the complexity of the task. This is because knowing the distribution $p_{Y|X}(y \mid x')$ for any one element $x'$ in each equivalence class is sufficient to predict $Y$. An extreme case of this would be the constant function which arguably has the simplest possible structure of any task $T_{X,Y}$. The equivalence class in this case is just $\mathcal{X}$ since $p_{Y|X}(y \mid x)$ is same for all $x \in \mathcal{X}$. Thus, the number of equivalence classes for the constant function is 1 (the minimum possible). This is expected since for a constant function there is no structure to learn from data whatsoever!

The following lemma (see Appendix A.1 for a proof) relates $C_Q(X;Y)$ and the number of equivalence classes, the latter being not easy to compute from finite samples.

**Proposition 1.** *Given a finite query set $Q$, $b$-valued query-answers $\{q(X)\}_{q \in Q}$ and any $\delta > 0$, the number of equivalence classes induced by the minimizer of* (2) *can be upper bounded by* $b^{C_Q(X;Y)+|Q|\sqrt{2\log(\frac{1}{\delta}))}}$ *with probability of misclassifying $X$ at most $\delta$.*

Proposition 1 indicates that for the same $Q$, tasks with greater $C_Q(X;Y)$ will have larger complexity (by comparing the upper bound on the number of equivalence classes). This also illustrates the role different bases play. For example, in visual recognition tasks if one queries intensities of all the pixels in the image at once then $C_Q(X;Y) = 1$ (Observing intensities at all the pixels is sufficient information to predict $Y$ from $X$). However $b$ in that case would be large (exponential in the size of the image). Instead if one queries individual pixels, $b$ would be the number of intensity values each pixel can take but $C_Q(X;Y) \geq 1$.

**Properties of $C_Q(X;Y)$.** The following proposition, whose proof can be found in Appendix A.2, establishes some key properties of our proposed measure of complexity.

**Proposition 2.** *For any query set $Q$ that is sufficient for task $T_{X,Y}$, $C_Q(X;Y)$ satisfies the following properties:*

1. *$C_Q(X;Y) \geq 0$. (non-negativity)*

2. *$C_Q(X;Y) = 0$ iff $X \perp\!\!\!\perp Y$. (trivial structure)*

3. *If $\forall x, x' \in \mathcal{X}, x \neq x', \exists y \in \mathcal{Y}$, such that $p_{Y|X=x}(y) \neq p_{Y|X=x'}(y)$, then $C_Q(X;Y) \geq C_Q(X;\tilde{Y})$ for all tasks $T_{X,\tilde{Y}}$ provided $Q$ is sufficient for $T_{X,\tilde{Y}}$. (total structure)*

4. *$C_Q(X;Y_1,Y_2) \leq C_Q(X;Y_1) + C_Q(X;Y_2)$ for any two tasks with $X \sim p_X(x)$ and $Y_1 \perp\!\!\!\perp Y_2 \mid X$. (sub-additivity under union)*

The property *"trivial structure"* captures the fact that if $Y$ is independent of $X$, the learning task is trivial (i.e., there is nothing to be learned about $Y$ from observing $X$ or functions of $X$).

The property *"total structure"* captures the intuition that if for a given task $T_{X;Y}$, the conditional distribution functioini $p_{Y|X=x}(y)$ is different $\forall x \in \mathcal{X}$, then learning is "impossible" (assume $Y$ is a categorical variable, which it is for most classification tasks) as no inductive bias would help in generalization to unseen inputs for such a task. For example, $Y = f(X)$ where $f : \mathcal{X} \to \mathcal{Y}$ is injective.

The last property *"sub-additivity under union"* is especially interesting in transfer learning settings where source task $T_{X;Y_1}$ and target task $T_{X;Y_2}$ have the same marginal for $X \sim p_X(x)$ but different conditionals $p_{Y_1|X}(y_1 \mid x)$ and $p_{Y_2|X}(y_2 \mid x)$. $C_Q(X;Y_1,Y_2)$ refers to the complexity of

task $T_{X;Y_1,Y_2}$, defined by $p_{XY_1Y_2}(x, y_1, y_2)$, where the corresponding sufficiency constraint in (2) becomes

$$p(y_1, y_2|x) = p(y_1, y_2|Code_Q^E(x)) \ \forall x \in \mathcal{X}, y_1 \in \mathcal{Y}_1, y_2 \in \mathcal{Y}_2. \tag{5}$$

This property could potentially be exploited to predict the success of transfer learning for different choices of source and target tasks. Further, the assumption $Y_1 \perp\!\!\!\perp Y_2 \mid X$ is not particularly strict as it simply implies given input $X$, knowledge of $Y_1$ is not required to predict $Y_2$ and vice-versa.

$\epsilon$-**approximate task complexity.** In practice, we are often interested in solving a task "approximately" rather than "exactly". To accommodate this, we extend the definition of Sufficiency in (2) to $\epsilon$-Approximate Sufficiency by relaxing it to $d\left(p(y \mid x), p(y \mid Code_Q^E(x))\right) \leq \epsilon \ \ \forall x \in \mathcal{X}$. Here, $d$ is any distance-like metric on distributions such as the KL-divergence, total variation, Wasserstein distance, etc. (Refer Appendix A.3).

Having established the above properties of our complexity measure, we note that unfortunately for any general query set $Q$, the problem in (2) is known to be NP-Complete and hence generally intractable (Laurent & Rivest, 1976). As a result, we instead consider a greedy approximation to $C_Q(X; Y)$ via an algorithm called Information Pursuit (IP) introduced by Geman & Jedynak (1996), which we describe in detail next.

## 3 APPROXIMATING TASK COMPLEXITY USING INFORMATION PURSUIT

From this section onwards we will assume $Q$ is a finite set. Information pursuit (IP) is a greedy algorithm introduced by Geman & Jedynak (1996) which provides an approximate solution to (2). The Encoder in IP, denoted as $E^{IP}$, is recursively defined as follows,

$$\begin{aligned}
q_1 &= E^{IP}(\emptyset) = \arg\max_{q \in Q} I(q(X); Y) \\
q_{k+1} &= E^{IP}(\{q_i, q_i(x)\}_{1:k})) = \arg\max_{q \in Q} I(q(X); Y|B_{x,k})
\end{aligned} \tag{6}$$

where $x$ is an input data-point and $I$ stands for mutual information. In other words, IP chooses the next query $q_{k+1}$ as the one whose answer maximizes the mutual information with $Y$ given the history of questions and answers about $x$ chosen by IP till time $k$, i.e., given the event $B_{x,k} := \{x' \in \mathcal{X} \mid \{q_i, q_i(x)\}_{1:k} = \{q_i, q_i(x')\}_{1:k}\}$. Ties in choosing $q_k$ are broken arbitrarily if the maximum is not unique. The algorithm stops when it satisfies the following condition:

$$q_{L+1} = q_{STOP} \quad \text{if} \quad \max_{q \in Q} I(q(X); Y|B_{x,m}) = 0 \ \forall m \in \{L, L+1, ..., L+T\}, \tag{7}$$

where $T > 0$ is a hyper-parameter chosen via cross-validation, with the rationale behind this choice provided in Appendix A.4. We will denote this sub-optimal solution $E^{IP}$ as $\tilde{C}_Q(X; Y)$. To compute an approximation to $C_Q^\epsilon(X; Y)$ we modify the stopping criteria in (7) to $\max_{q \in Q} I(q(X); Y|B_{x,m}) \leq \epsilon$ and call this estimate $\tilde{C}_Q^\epsilon(X; Y)$.

### 3.1 APPROXIMATION GUARANTEES FOR INFORMATION PURSUIT

While in general it is difficult to have any performance guarantees of IP, in the specialized setting in which $Q$ indexes the set of all possible binary functions of $X$, such that $H(q(X) \mid Y) = 0 \ \forall q \in Q$ and $Y$ is a function of $X$, we have the following proposition (see Appendix A.6 for a proof).

**Proposition 3.** *Given task $T_{X;Y}$ with $Y$ being a discrete random variable. If there exists a function $f$ such that $Y = f(X)$ and $Q$ is the set to all possible binary functions of $X$ such that $H(q(X) \mid Y) = 0 \ \forall q \in Q$ then $H(Y) \leq \tilde{C}_Q(X; Y) \leq H(Y) + 1$.*

While the above proposition is often considered to be true, this is the first time a rigorous proof has been presented (to the best of our knowledge). Thus, in this special case, from (3) we have that $\tilde{C}_Q(X; Y) \leq C_Q(X; Y) + 1$ and thus the IP algorithm will be a tight approximation to our proposed complexity measure.

### 3.2 Information Pursuit using Variational Autoencoders + Normalizing Flows

IP requires probabilistic models relating query-answers and data to compute the required mutual information terms in (6). Specifically, computing $q_{k+1}$ in (6) (for any iteration number $k$) requires computing the mutual information between $q(X)$ and $Y$ given the history $B_{x,k}$ till time $k$. As histories become longer, we quickly run out of samples in our dataset which belong to the event $B_{x,k}$. As a result, non-parametric sample-based methods to estimate mutual information (such as Belghazi et al. (2018)) would be impractical. In this subsection, we propose a model-based approach to address this challenge for a general task $T_{X;Y}$ and query set $Q$. In section 4 we adapt this model to the specific case where $Q$ indexes image patches.

**Information Pursuit Generative Model.** Let $Q(X) = \{q(X) : q \in Q\}$. To make learning tractable, we introduce latent variables $Z$ to account for all the dependencies between different query answers and posit the following factorization of $Q(X), Y, Z$

$$p_{Q(X)ZY}(Q(x), \eta, y) = \prod_{q \in Q} p_{q(X)|ZY}(q(x) \mid \eta, y) p_Y(y) p_Z(\eta). \tag{8}$$

Throughout the paper $\eta$ and $q(x)$ denote the realizations of $Z$ and $q(X)$ respectively. Equation (8) is a reasonable assumption unless the answers $q(X)$ are causally related to each other, (Reichenbach's common cause principle (Hofer-Szabó et al., 1999)).

In other words, assuming that the query-answers are conditionally independent given the hypothesis and "some" latent vector is benign and ubiquitous in many machine learning applications.

1. **$q(X)$ as object presence indicators evaluated at non-overlapping windows:** Let $Q$ be a set of non-overlapping windows in the image $X$ with $q(X)$ as the random variable indicating the presence of an object at the $q^{th}$ location. The correlation between the $q$s is entirely due to latent image generating factors $Z$, such as lighting, camera position, scene layout, and texture along with the scene description signal $Y$.

2. **$q(X)$ as snippets of speech utterances:** A common assumption in speech recognition tasks is that the audio frame features ($q(X)$) are conditionally independent given latent phonemes $Z$ (which is often modelled as a Hidden Markov Model).

This latent space $Z$ is often a lower-dimensional space compared to the original high-dimensional $X$. We learn $Z$ from data in an unsupervised manner using variational inference. Note, this assumption of conditional independence will not hold in scenarios where $q(X)$ directly cause each other, for instance, if in example 1, we considered overlapping patches.

Specifically, we parameterize the distributions $\{p_\omega(q(x) \mid \eta, y) \; \forall q \in Q\}$ with a neural network with shared weights $\omega$ and call it the ***Decoder Network***. These weights are learnt using stochastic Variational Bayes (Kingma & Welling, 2013) by introducing an approximate posterior distribution $p'_\phi(\eta \mid y, Q(x))$ parameterized by another neural network with weights $\phi$ called the ***Encoder Network*** and priors $p_Y(y)$ and $p_Z(\eta)$.

**Implementing $E_Q^{IP}$ using the generative model.** Using the learnt Decoder network one can estimate the distribution $p_{q(X)Y}(q(x), y)$ via Monte Carlo Integration and compute $q_1 = E^{IP}(\emptyset)$. For subsequent queries ($k > 1$), the computation of $q_{k+1}$ requires the mutual information conditioned on current history $B_{x,k}$, which can be calculated from the distribution

$$p(q(x), y \mid B_{x,k}) = \int p(q(x) \mid \eta, y) p(\eta \mid y, B_{x,k}) p(y \mid B_{x,k}) d\eta. \tag{9}$$

To estimate the left-hand side of (9) via Monte Carlo integration, one needs to sample $N$ i.i.d. samples $\eta^i \sim p(\eta \mid y, B_{x,k})$ and compute $\frac{1}{N} \sum_{i=1}^{N} p_\omega(q(x) \mid \eta^{(i)}, y) p(y \mid B_{x,k})$, where the term $p(y \mid B_{x,k})$ can be estimated recursively with $p(y \mid B_{x,0}) := p_Y(y)$. Appendix A.7 gives more details on these computations. Note, however, it is not straightforward to sample from this posterior $p(\eta \mid y, B_{x,k})$ without resorting to advanced Markov Chain Monte Carlo sampling techniques, which often suffers from the curse of dimensionality and is computationally intensive. To mitigate this issue we implement $p(\eta \mid y, B_{x,k})$ by another neural network trained to learn this posterior.

**Estimating $p(\eta \mid y, B_{x,k})$ with Normalizing Flows.** We amortize the cost of modelling $p(\eta \mid y, B_{x,k})$ for each history encountered during the $E^{IP}$ recursion by assuming the existence of a function $\Psi$ such that $p(\eta \mid y, B_{x,k}) = \Psi(\{(q_i, q_i(x))\}_{1:k}, y, \eta) \; \forall x \in \mathcal{X}$ for any iteration $k$.

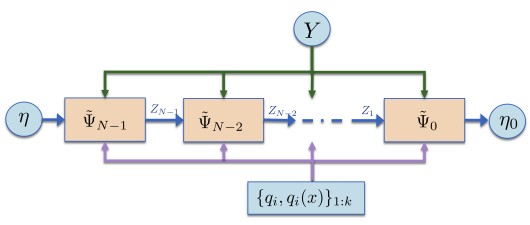

We employ Normalizing Flows (Dinh et al., 2016) to approximate $\Psi$ from data. Specifically, we construct a sequence of invertible mappings of the form $\eta_l = \tilde{\Psi}_l(\{(q_i, q_i(x))\}_{1:k}, y, \eta_{l+1}; \gamma_l)$, each parameterized by a neural network with weights $\gamma_l$, such that $\eta_0$ is constrained to be normally distributed. The composite function $\eta_0 = \tilde{\Psi}(\{(q_i, q_i(x))\}_{1:k}, y, \eta; \gamma)$ is a composition of $N$ neural networks with parameters $\gamma = \{\gamma_l\}_{l \in \{1,2,...,N\}}$. We call this network,

Figure 2: *Conditional Inference Network.*

$\tilde{\Psi}$, as the ***Conditional Inference Network*** (Refer Fig. 2). Refer Appendix A.8 for details on the training procedure.

Using the trained $\tilde{\Psi}$ one can sample from the posterior $p(\eta \mid y, B_{x,k})$, as required for (9) for any observed history $B_{x,k}$. The sampling procedure is as follows: (i) Sample $\eta_0 \sim \mathcal{N}(0, I_d)$ (assuming $\eta_0 \in \mathbb{R}^d$); (ii) Compute $\eta = \tilde{\Psi}^{-1}(\{(q_i, q_i(x))\}_{1:k}, y, \eta_0)$.

# 4 CASE STUDY: COMPLEXITY OF BINARY IMAGE CLASSIFICATION TASKS

As a practical instantiation of our theory, we concentrate on the task of classifying binary images. We choose $Q$ as the set of image patches with answers being the intensities observed at the patch. The reason for this choice is two-fold: (i) Patches provide a sufficiently rich query set to compare different learning tasks on binary images and allow us to measure the *task-specific input feature complexity* of different tasks; (ii) From a practical stand-point, state-of-the-art deep generative models for binary images can be assumed to be "perfect" models allowing us to illustrate the usefulness of the framework with minimal modelling bias.

For all our experiments, we considered $Q$ as index set to all $3 \times 3$ overlapping patches in the image. This requires some modelling changes. Recall, (8) only holds if $q(X)$ are not causally related. In case of overlapping patches, this assumption is clearly violated. So instead we model (8) at the level of pixels $X_j$ ($X$ denoting the binary image, and $j$ the $j^{th}$ pixel), $p(x_j, \eta, y) = \prod_{x_j \in X} p(x_j \mid \eta, y) p(y) p(\eta)$. Further network training details in Appendix A.9.

**Complexity increases with decrease in signal-to-noise ratio.** We tested the effect of two different task-specific nuisances on $\tilde{C}_Q^\epsilon(X; Y)$ for MNIST classification by, (i) ***MNIST-$\alpha$:*** randomly flipping pixels in MNIST images with probability $\alpha \in [0, 1]$; (ii) ***MNIST-Translated:*** translating the digits by at most $4$ pixels. Fig. 3a shows the results. The plot shows the trade-off between accuracy and task complexity for different values of $\epsilon$. To normalize for the effect of different datasets having different Bayes error rates, we report the trade-off using relative test accuracies which are the accuracies obtained by predicting $Y$ according to $\arg\max_{y \in \mathcal{Y}} p(y \mid B_{x,L})^2$ divided by the prediction made upon observing the entire image (all the patches indexed by $Q$). The results indicate that for almost all desired relative accuracy levels, MNIST, MNIST-0.05, and MNIST-0.1 are in increasing level of complexity. Our experiments also indicate that the complexity of non-centered MNIST digit classification is the greatest (evaluated at any fixed accuracy level).

**Comparing different classification tasks.** While in the previous experiment, there was an expected "correct" trend (complexity increases as nuissance level increases), the complexity ordering is not so clear intuitively when comparing across datasets. We evaluated our framework to compute complexities of image classification on four different datasets of binary images, namely, MNIST, FashionMNIST, KMNIST, and Caltech Silhouettes. Fig. 3b reports the results. Our findings indicate that MNIST $<$ FashionMNIST $<$ KMNIST $<$ Caltech Silhoettes in terms of task complexity at almost all relative test accuracy levels.

---

[2]Recall L is the iteration after which IP terminates

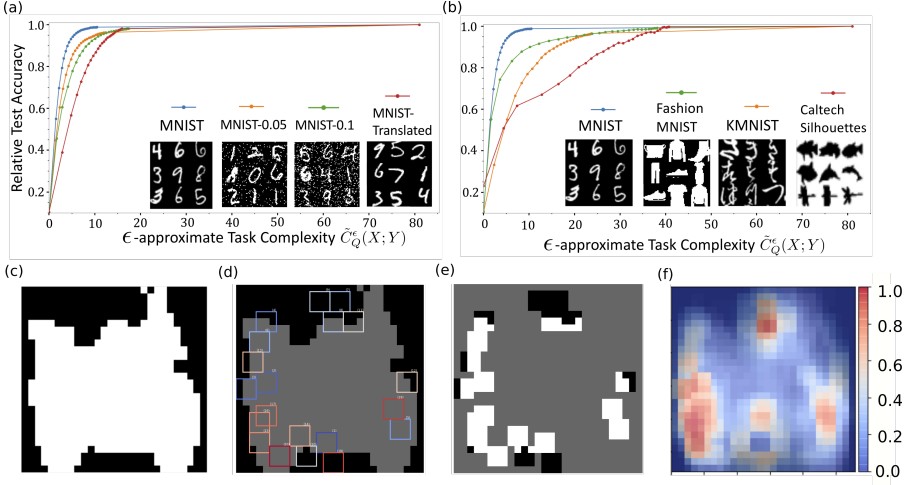

Figure 3: *The plots (a) & (b) show the trade-off between accuracy and complexity as $\epsilon$ is varied (c), (d), (e) & (f) pertain to our discussion on interpretability.* (a) *Complexity results on MNIST with different levels of nuissances;* (b) *Complexity results for different image classification tasks;* (c) *test image $x^0$ from Caltech Silhouettes dataset with class "Motorbike";* (d) *patches queried by $E^{IP}$ before termination for $x^0$ (shown by the overlayed coloured $3 \times 3$ boxes);* (e) *Part of image $x^0$ observed through the queried patches;* (f) *Heatmap for probability a pixel would be visited by the IP Encoder for a randomly chosen image with label "Motorbike" from the Caltech Silhouettes dataset.*

**Connections with interpretability.** A common theme in all the different notions of interpretability in ML literature is a "short description length". An interesting consequence of our formulation is that when evaluated using queries that index image patches, it gives a principled way of selecting the most important parts of an image for a task. Fig. 3c, d, e, & f illustrate this with an example. IP predicts the image label is "motorbike" by just observing the edges of the silhouette which intuitively should be the most important parts of the image for this task. Indeed, Fig. 3f reveals that $E^{IP}$ visits these edges with high probability for any random image that has label "Motorbike". This heatmap makes sense since the data set only has centered images, for the general case a more in-depth analysis would be required. This property of IP could be utilized to develop more interpretable ML algorithms.

## 5 CONCLUSION AND RELATED WORK

In this paper, we introduced a novel notion of task complexity intimately tied to a query set $Q$. In the following paragraphs, we will briefly discuss some relevant prior work with connections to our proposed measure.

The information pursuit algorithm has roots in the seminal work of (Geman & Jedynak, 1996), which proposes an active testing framework for tracking roads in an image. That algorithm was extended in (Sznitman & Jedynak, 2010) for face detection and localization, in Sznitman et al. (2012) for detection and tracking of surgical instruments, and in (Jahangiri et al., 2017) for scene interpretation. Also, while (Sznitman & Jedynak, 2010; Jahangiri et al., 2017) learn generative models for their tasks, their models are radically different from ours.

The problem of classifying objects by sequentially observing different image locations has been recently re-branded as *Hard Attention* in vision (Mnih et al., 2014; Elsayed et al., 2019; Li et al., 2016) and several deep learning solutions have been proposed. These methods typically try to learn a policy for iteratively choosing different parts of an image for solving different visual tasks. Optimization techniques from reinforcement learning are often employed to achieve this. High variance in the gradient estimates and scalability issues prevent their widespread adoption. It would be interesting to see if IP can be combined with reinforcement learning-based approaches to design better reward functions to facilitate efficient policy search.

The *information bottleneck* (IB) method proposed by Tishby et al. (2000) is perhaps the closest to our work. They define complexity in terms of the mutual information (MI) between input $X$ and its representations $\tilde{X}$ such that $\tilde{X}$ preserves a certain amount (determined by a user-defined parameter) of information about the output variable $Y$. In a way, their measure of complexity accounts for the relationship between $X$ and $Y$. However, this complexity isn't very useful for comparing different learning tasks. For instance, in Fig. 1 & 2 in Kolchinsky et al. (2019) the MI between $X$ and $\tilde{X}$ for FashionMNIST and MNIST datasets are roughly the same for the same level of accuracy. This is a problem since we know from practical experience that MNIST is a much "simpler" than Fashion-MNIST. Interestingly, when $Q$ is taken to be the set all possible binary functions of $X$ our proposed measure recovers IB and its variants (Strouse & Schwab, 2017). That discussion however is out of the scope of this paper. We note in passing that there has recently been work on quantifying information of a system under limited computation and model constraints (Xu et al., 2020) which could potentially be explored in the future in conjunction with our framework. In a sense, the choice of $Q$ constrains the way information from $X$ can be extracted to predict $Y$.

Task complexity measures based on Kolmogorov Complexity are not computable (Li, 2006; Achille et al., 2019b; Vereshchagin & Vitányi, 2004). These measures are based on the idea of finding the minimum length computer program that given input $x$ outputs label $y$ for every $(x, y)$ in the training dataset. This is in stark contrast to our definition which is a property of the joint distribution $p_{XY}(x, y)$ and not any given finite dataset. Computing $C_Q(X; Y)$ is NP-Complete but not uncomputable. The implication of this is that there exist dynamic programming based solutions that *exactly compute* $C_Q(X; Y)$. The complexity of these algorithms are typically exponential in $|Q|$ and so feasible only when $|Q|$ is small. For large $|Q|$, we must turn to approximations and Information Pursuit is one such strategy. On the other hand, an algorithm for computing Kolmogorov Complexity does not exist, let alone an efficient one. Besides computability, the more pressing issues with Kolmogorov complexity is that the measures are sensitive to permutations of the dataset which is undesirable. Secondly, Kolmogorov complexity fails to distinguish between memorization and learning. A dataset sampled from $p_{XY}(x, y)$ where $Y$ is independent of $X$ will have the maximum Kolmogorov-based complexity measure. However, from a learning point of view there is nothing to learn - an optimal strategy is to simply predict $p(Y \mid X) = p(Y)$ regardless of the value of $X$! So, the task complexity of such tasks should be 0. Achille et al. (2019b) presents for a more detailed discussion on this. The proposed measure $C_Q(X; Y)$ is not dataset permutation-sensitive since it is a property of the distribution $p_{XY}(x, y)$. Moreover, $C_Q(X; Y) = 0$ when $Y$ is independent of $X$ (See Proposition 2.2) and so distinguishes learning from memorization.

Our work is related in spirit to the work of Achille et al. (2019b) which introduces an alternate measure of task complexity based on the intuition that the information stored in the weights of a trained network can be used as a measure of task complexity. They show that their measure recovers Kolmogorov's complexity, Shannon's information, Fisher's information as special cases, and is also related to PAC-Bayes generalization bounds.

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

# A    APPENDIX

In proofs of propositions and lemmas we rewrite the statement for convenience.

## A.1    PROOF OF PROPOSITION 1

**Proposition.** *Given a finite query set Q, b-valued query-answers $\{q(X)\}_{q \in Q}$ and any $\delta > 0$, the number of equivalence classes induced by the minimizer of (2) can be upper bounded by $b^{C_Q(X;Y) + |Q|\sqrt{2\log(\frac{1}{\delta})})}$ with probability of misclassifying $X$ at most $\delta$.*

*Proof.* Let $d_Q(X)$ be a random variable denoting the optimal code-length for $X$ using $E^*$, the solution to (2). Since, $0 \le d_Q(X) \le |Q|$. From the definition, $C_Q(X;Y) = \mathbb{E}_X [d_Q(X)]$. So, Hoeffding's Lemma we have

$$d_Q(X) - C_Q(X;Y) \sim subG(\frac{|Q|^2}{4}) \tag{10}$$

Since $d_Q(X)$ is sub-gaussian, for any $\delta > 0$

$$\mathbb{P}\left(d_Q(X) > C_Q(X;Y) + |Q|\sqrt{2\log\frac{1}{\delta}}\right) \le \delta \tag{11}$$

Notice, the prefix-free nature of $E^*$ naturally gives rise to a tree with $d(X)$ being the depth of the corresponding tree $T_Q$ for input $X$. The number of equivalence classes is exactly the number of leaves of $T_Q$. Given, each query-answer $q(X)$ is b-valued. So we can upper bound the number of leaves by replacing $T_Q$ with a balanced $b$-ary tree with $b^{C_Q(X;Y) + |Q|\sqrt{2\log\frac{1}{\delta}}}$ leaves called $\tilde{T}_Q$. $\tilde{T}_Q$ can be constructed from $T_Q$ as follows:

1. For any leaf node with depth $d(X) < C_Q(X;Y) + |Q|\sqrt{2\log\frac{1}{\delta}}$ vacuously increase the depth till equality is achieved by repeatedly asking the last query. For any internal node in $T_Q$ that is not balanced, add vacuous subtrees by randomly selecting queries from $Q$ (no $x \in \mathcal{X}$ would be sent to these new subtrees) to make the tree balanced with each internal node having $b$ children nodes. As a result, $\forall x \in \mathcal{X}$ with $d(x) < C_Q(X;Y) + |Q|\sqrt{2\log\frac{1}{\delta}}$, $\tilde{T}_Q(X)$ would compute the same posterior as $T_Q(X)$, that is $p(y \mid Code_Q^{E^*}(x))$, and thus not make an error.

2. For any leaf node with depth $d(X) > C_Q(X;Y) + |Q|\sqrt{2\log\frac{1}{\delta}}$, cut the path at depth $u(X) = C_Q(X;Y) + |Q|\sqrt{2\log\frac{1}{\delta}}$ by merging the subtree rooted at depth $u(X)$ and creating a leaf node. An input $x \in \mathcal{X}$ sent to this new leaf node would be misclassified. The probability of such an event is given by (11).

$\square$

## A.2    PROOF OF PROPOSITION 2

**Proposition.** *For any query set Q that is sufficient for task $T_{X,Y}$, $C_Q(X;Y)$ satisfies the following properties.*

1. *$C_Q(X;Y) \ge 0$. (non-negativity)*

2. *$C_Q(X;Y) = 0$ iff $X \perp\!\!\!\perp Y$. (trivial structure)*

3. *If $\forall x, x' \in \mathcal{X}, x \ne x', \exists y \in \mathcal{Y}$, such that $p_{Y|X=x}(y) \ne p_{Y|X=x'}(y)$, then $C_Q(X;Y) \ge C_Q(X;\tilde{Y})$ for all tasks $T_{X,\tilde{Y}}$ provided $Q$ is sufficient for $T_{X,\tilde{Y}}$. (total structure)*

4. $C_Q(X; Y_1, Y_2) \leq C_Q(X; Y_1) + C_Q(X; Y_2)$ *for any two tasks with* $X \sim p_X(x)$ *and* $Y_1 \perp\!\!\!\perp Y_2 \mid X$. *(sub-additivity under union)*

*Proof.*

1. Follows trivially from the definition.

2. For proving the "if" part, observe that if $X \perp\!\!\!\perp Y$ then $p_{Y|X=x}(y) = p_Y(y) \, \forall y \in \mathcal{Y}, x \in \mathcal{X}$. Choose $E_1$ to be an encoder such that $q_1 = q_{STOP}$. Then

$$\mathbb{E}_X \left( |Code_Q^{E_1}(X)| \right) = 0$$

$E_1$ is trivially prefix-free (all $x \in \mathcal{X}$ gets mapped to the same code $\emptyset$). Recall that $p(y|Code_Q^{E_1}(x))$ should be interpreted as the conditional probability of $y$ given the event $\{x' \in \mathcal{X} \mid Code_Q^{E_1}(x) = Code_Q^{E_1}(x')\}$. Since $\forall x \in \mathcal{X}, Code_Q^{E_1}(x) = \emptyset$, $p(y \mid Code_Q^{E_1}(x)) = p(y \mid \mathcal{X}) = p(y)$. Hence, $E_1$ also provides a sufficient code and thus is a feasible solution to the optimization problem in (2). From property 1 we know that $E_1$ achieves the optimal $C_Q(X; Y)$.
The proof for the "only if" part is very similar. Consider $E^*$ as the optimal solution to (2) and $C_Q(X; Y) = 0$. This implies $Code_Q^{E^*}(x) = \emptyset \, \forall x \in \mathcal{X}$. Since $E^*$ is sufficient, $p(y \mid x) = p(y \mid Code_Q^{E^*}(x)) = p(y) \, \forall x \in \mathcal{X}, y \in \mathcal{Y}$. This implies $X \perp\!\!\!\perp Y$.

3. Let $E_T$ denote the minimizer of (2) for the task $T_{X;Y}$ where $Y$ given $X$ is distributed as stated. From sufficiency we know that no two different inputs $x, x' \in \mathcal{X}$ could have the same code, that is, $Code_Q^{E_T}(x) \neq Code_Q^{E_T}(x') \, \forall x, x' \in \mathcal{X}, x \neq x'$. Consider any other task $T_{X, \tilde{Y}}$ defined by $p(X, \tilde{Y})$ with the same marginals $X \sim p_X(x)$ but possibly different conditional $\tilde{Y} \sim p_{\tilde{Y}|X}(\tilde{y} \mid x)$ such that $Q$ is also sufficient for $T_{X, \tilde{Y}}$. $E_T$ is a feasible solution for (2) with respect to task $T_{X, \tilde{Y}}$ since $p(\tilde{y} \mid Code_Q^{E_T}(x)) = p(\tilde{y} \mid \{x\})$. Thus, $C_Q(X; \tilde{Y}) \leq \mathbb{E}_X \left( |Code_Q^{E_T}(X)| \right)$ for any other task $T_{X; \tilde{Y}}$ for which $Q$ is sufficient.

4. Let $E_1$ and $E_2$ be the optimal encoders for tasks $T_{X;Y_1}$ and $T_{X;Y_2}$ respectively. Construct an encoder $E_{12}$ for that task $T_{X;Y_1,Y_2}$ by concatenating the two codes. It is not hard to see that

$$\mathbb{E}_X \left( |Code_Q^{E_{12}}(X)| \right) \leq \mathbb{E}_X \left( |Code_Q^{E_1}(X)| \right) + \mathbb{E}_X \left( |Code_Q^{E_2}(X)| \right).$$

The inequality is due to the fact that (query,answer) tuples can overlap in the codes constructed by two encoders $E_1$ and $E_2$ for the same input $x$. $E_{12}$ is prefix-free by construction. Now, for any observation $y_1$ of output $Y_1$

$p(y_1 \mid Code_Q^{E_{12}}(x))$
$= p(y_1 \mid Code_Q^{E_1}(x) \cup Code_Q^{E_2}(x))$
$= p(y_1 \mid \{x' \in \mathcal{X} : Code_Q^{E_1}(x') \cup Code_Q^{E_2}(x') = Code_Q^{E_1}(x) \cup Code_Q^{E_2}(x)\})$
$= p(y_1 \mid \{x' \in \mathcal{X} : Code_Q^{E_1}(x') = Code_Q^{E_1}(x)\} \cap \{x' \in \mathcal{X} : Code_Q^{E_2}(x') = Code_Q^{E_2}(x)\})$
$= p(y_1 \mid x)$

The last equality appeals to the fact that $E_1$ satisfies the "sufficiency" constraint in (2) for task $T_{X;Y_1}$. Similarly, $p(y_2 \mid Code_Q^{E_{12}}(x)) = p(y_2 \mid x)$ for any observation $y_2$ of output $Y_2$. Given, $Y_1 \perp\!\!\!\perp Y_2 \mid X$. This implies $\forall x \in \mathcal{X}$

$$p(y_1, y_2 \mid x) = p(y_1 \mid x)p(y_2 \mid x)$$
$$= p(y_1 \mid Code_Q^{E_{12}}(x))p(y_2 \mid Code_Q^{E_{12}}(x))$$
$$= p(y_1, y_2 \mid Code_Q^{E_{12}}(x))$$

This proves $E_{12}$ is sufficient for the task $T(X; Y_1, Y_2)$. Since $E_{12}$ is a feasible solution for the optimization problem with respect to $T(X; Y_1, Y_2)$ we get the required inequality.

$\square$

## A.3 $\epsilon$-APPROXIMATE COMPLEXITY OF TASK

In practice, we are often interested in solving a task "approximately" rather than "exactly". This requires introducing a notion of approximate sufficiency instead of exact sufficiency, and we extend the definition in (2) to incorporate this.

$$C_Q^\epsilon(X; Y) := \min_E \mathbb{E}_X \left[ |Code_Q^E(X)| \right] \tag{12}$$

$$\text{s.t.} \quad d\left(p(y \mid x), p(y \mid Code_Q^E(x))\right) \leq \epsilon \ \forall x \in \mathcal{X} \quad \text{(Approx. Sufficiency)} \tag{13}$$

Here, $d$ is any distance-like metric on distributions such as the KL-divergence, total variation, Wasserstein distance, etc. Additionally, if $d$ is convex in both its arguments, symmetric and satisfies the triangle-inequality then $C_Q^\epsilon(X; Y)$ satisfies the properties in Proposition 2 with two key differences:

1. In Property 3, $C_Q^\epsilon(X; Y)$ is the complexity of the task for which $d\left(p_{Y|X=x}(y), p_{Y|X=x'}(y)\right) > 2\epsilon \ \forall x, x' \in \mathcal{X}, x \neq x'$.

2. In Property 4, the relation is $C_Q^\epsilon(X; Y_1, Y_2) \leq C_Q^{\frac{\epsilon}{2}}(X; Y_1) + C_Q^{\frac{\epsilon}{2}}(X; Y_2)$.

## A.4 TERMINATION CRITERIA FOR $E_Q^{IP}$

Ideally we would like to terminate ($E^{IP}$ outputs $q_{STOP}$) after $L$ steps if

$$p(y \mid x) = p(y \mid x') \ \forall x, x' \in B_{x,L}, \ y \in \mathcal{Y} \tag{14}$$

However, detecting this is difficult in practice. We have the following lemma.

**Lemma A.4.1.** *Assume $Y \perp\!\!\!\perp q(X) \mid X \ \forall q \in Q$. If event $B_{x,L}$ satisfies the condition specified by (14) then for all subsequent queries $q_m$, $m \geq L$, $\max_{q \in Q} I(q(X); Y | B_{x,m}) = 0$. Since ties are broken arbitrarily if the maximum not unique, $E^{IP}$ chooses any $q \in Q$ as a subsequent query $q_m$.*

Refer to Appendix A.5 for a proof. The assumption $Y \perp\!\!\!\perp q(Xx) \mid X \ \forall q \in Q$ is generally true since we have the following Markov Chain $Y \to X \to q(X) \ \forall q \in Q$.

Using Lemma A.4.1, the correct stopping criteria should be

$$L = \inf\{k \in \{1, 2, ..., |Q|\} : \max_{q \in Q} I(q(X); Y | B_{x,m}) = 0 \ \forall m \geq k, m \leq |Q|\} \tag{15}$$

Evaluating (15) would be computationally costly since it would involve processing all the queries for every input $x$. We employ a more practically amenable criteria

$$q_{L+1} = q_{STOP} \quad \text{if} \quad \max_{q \in Q} I(q(X); Y | B_{x,m}) = 0 \ \forall m \in \{L, L+1, ..., L+T\} \tag{16}$$

$T > 0$ is a hyper-parameter chosen via cross-validation. Note, it is possible that there does not exist any informative query in one iteration, but upon choosing a question there suddenly appears informative queries in the next iteration. For example, consider the XOR problem. $X \in \mathbb{R}^2$ and $Y \in \{0, 1\}$. Let $Q$ be the set to two axis-aligned half-spaces. Both half-spaces have zero mutual information with $Y$. However, upon choosing any one as $q_1$, the other half-space is suddenly informative about $Y$.

## A.5 PROOF OF LEMMA A.4.1

**Lemma.** *Assume $Y \perp\!\!\!\perp q(X) \mid X \ \forall q \in Q$. If event $B_{x,L}$ satisfies the condition specified by (14) then for all subsequent queries $q_m$, $m \geq L$, $\max_{q \in Q} I(q(X); Y | B_{x,m}) = 0$. Since ties are broken arbitrarily, $E^{IP}$ chooses any $q \in Q$ as a subsequent query $q_m$.*

*Proof.* Recall each query $q$ partitions the set $\mathcal{X}$ and $B_{x,L}$ is the event $\{x' \in \mathcal{X} \mid \{q_i, q_i(x)\}_{1:L} = \{q_i, q_i(x')\}_{1:L}\}$. It is easy to see that if $B_{x,L}$ satisfies the condition specified by (14) then

$$P(y \mid B_{x,m}) = P(y \mid x') \, \forall x' \in B_{x,m} \, \forall m \geq L, \, \forall q \in Q \tag{17}$$

This is because subsequent query-answers partition a set in which all the data points have the same posterior distributions[3]. Now, $\forall q \in Q, \, \forall a \in Range(q), \, y \in \mathcal{Y}$

$$p(q(X) = a, y | B_{x,m}) = p(q(X) = a \mid B_{x,m}) p(y \mid q(X) = a, B_{x,m}) \tag{18}$$

eq: chain rule of prob is just an application of the chain rule of probability. The randomness in $A_q(X)$ is entirely due to the randomness in $X$. For any $a \in Range(A_q), y \in \mathcal{Y}$

$$\begin{aligned}
p(y \mid q(X) = a, B_{x,m}) &= \sum_{x' \in B_{x,m} \cap \{x \in \mathcal{X} | q(X) = a\}} p(y, x' \mid a, B_{x,m}) \\
&= \sum_{x' \in B_{x,m} \cap \{x \in \mathcal{X} | q(X) = a\}} p(y \mid x', a, B_{x,m}) p(x' \mid a, B_{x,m}) \\
&= \sum_{x' \in B_{x,m} \cap \{x \in \mathcal{X} | q(X) = a\}} p(y \mid x') p(x' \mid a, B_{x,m}) \\
&= p(y \mid B_{x,m}) \sum_{x' \in B_{x,m} \cap \{x \in \mathcal{X} | q(X) = a\}} p(x' \mid a, B_{x,m}) \\
&= p(y \mid B_{x,m})
\end{aligned} \tag{19}$$

The first equality is an application of the law of total probability, third due to conditional independence of the history and the hypothesis given $X = x'$ (assumption) and the fourth by invoking ((17)).

Substituting (19) in (18) we obtain $Y \perp\!\!\!\perp q(X) \mid B_{x,m} \, \forall m \geq L, q \in Q$. This implies that for all subsequent queries $q_m, m > L, \max_{q \in Q} I(q(X); Y | B_{x,m}) = 0$. Hence, Proved.

$\square$

### A.6 Proof of Proposition 3

**Proposition.** *Given task $T_{X;Y}$ with $Y$ being a discrete random variable. If there exists a function $f$ such that $Y = f(X)$ and $Q$ is the index set to all possible binary functions of $X$ such that $H(q(X) \mid Y) = 0 \, \forall q \in Q$ then $H(Y) \leq \tilde{C}_Q(X; Y) \leq H(Y) + 1$.*

We make two remarks before turning to the proof.
**Remark 1:**

The task is to determine the true state of a latent variable $Y \in \mathcal{Y}$ based on querying an observed data point $x^0$. We assume $Y = f(X)$ with $f$ unknown. Were $Y$ observable, the natural queries would be indexed by subsets $D \subset \mathcal{Y}$, one query for every $D \subset \mathcal{Y}$, namely $q(Y) = 1$ if $Y \in D$ and 0 otherwise. (This is essentially the classic "twenty questions game", but with an "oracle" and "complete tests".) There are $2^{|\mathcal{Y}|}$ such queries and obviously they collectively determine $Y$. Now since $Y = f(X)$, these queries are, at least implicitly, functions of the data $X$, but we need *realizable* functions, not requiring knowledge of $f$. So our fundamental assumption is that for each subset $D \in \mathcal{Y}$ the corresponding subset $D' \in \mathcal{X}$ ($D' = f^{-1}(D)$) can be checked for inclusion of $X$, i.e., $Y \in D$ if and only if $X \in D'$. Or, what is the same, a binary query $q(X)$ (and still denoted $q$ for simplicity) with $q(X) = q(Y)$. In effect, we are assuming that whereas we cannot determine $Y$ directly from $X$, we can answer simple binary queries which determine $Y$ *and* can be expressed as observable data features.

**Remark 2:** The sequence of queries $q_1, q_2, \ldots$ generated by the IP algorithm for a particular data point can be seen as one branch, root to leaf, of a decision tree constructed by the standard machine learning strategy based on successive reduction of uncertainty as measured by entropy:

---

[3]We refer to the distribution $p(y \mid x)$ for any $x \in \mathcal{X}$ as the posterior distribution of $x$.

$q_1 = \arg\max_{q \in Q} I(q(X); Y), q_{k+1} = \arg\max_{q \in Q} I(A_q(X); Y|B_{x^0,k})$ where the $B_{x^0,k}$ is the event that for the first $k$ questions the answers agree with those for $x^0$. We stop as soon as $Y$ is determined. Whereas a decision tree accommodates all $x$ simultaneously, the questions along the branch depends on having a particular, fixed data point. But the learning problem in the branch version ("active testing") is exponentially simpler.

**Proof of Proposition 3.1:** The lower bound $H(Y) \leq \tilde{C}_Q(X; Y)$ comes from Shannon's source coding theorem for symbol codes.

Now for the upper bound, since $I(q(X); Y|B_{x^0,k}) = H(q(X)|B_{x^0,k}) - H(q(X)|Y, B_{x^0,k})$ and since $Y$ determines $q(Y)$ and hence also $q(X)$, the second entropy term is zero (since given $H(A_q(X) \mid Y) = 0$). So our problem is maximize the conditional entropy of the binary random variable $q(X)$ given $B_{x^0,k}$. So the IP algorithm is clearly just "divide and conquer":

$$q_1 = \arg\max_{q \in Q} H(q(X)),$$

$$q_{k+1} = \arg\max_{q \in Q} H(q(X)|B_{x^0,k}).$$

Equivalently, since entropy of a binary random variable $\rho$ is maximized when $P(\rho) = \frac{1}{2}$,

$$q_{k+1} = \arg\min_{q \in Q} |P(q(X) = 1|B_{x^0,k}) - \frac{1}{2}|.$$

Let $\mathcal{Y}_k$ be the set of "active hypotheses" after $k$ queries (denoted as $\mathcal{A}_k$), namely those $y$ with positive posterior probability: $P(Y = y|B_{x^0,k}) > 0$. Indeed,

$$P(Y = y|B_{x^0,k}) = \frac{P(B_{x^0,k}|Y = y)p(y)}{\sum_y P(B_{x^0,k}|Y = y)p(y)}$$

$$= \frac{1_{\mathcal{Y}_k}p(k)}{\sum_{y \in \mathcal{A}_k} p(y)}$$

since

$$P(B_{x^0,k}|Y = y) = \begin{cases} 1, & \text{if } y \in \mathcal{A}_k \\ 0, & y \notin \mathcal{A}_k \end{cases}$$

In particular, the classes in the active set have the *same relative weights* as at the outset. In summary:

$$p(y|B_{x^0,k}) = \begin{cases} p(y)/\sum_{\mathcal{A}_k} p(l), & y \in \mathcal{A}_k \\ 0, & \text{otherwise} \end{cases}$$

The key observation to prove the theorem is that if a hypothesis $y$ generates the same answers to the first $m$ or more questions as $y^0$, and hence is active at step $m$, then its prior likelihood $p(y)$ is at most $2^{-(m-1)}$, $m = 1, 2, \ldots$. This is intuitively clear: if $y$ has the same answer as $y^0$ on the first question, and $p(y^0) > \frac{1}{2}$, then only one question is needed and the active set is empty at step two; if $q_1(y) = q_1(y^0)$ and $q_2(y) = q_2(y^0)$ and $p(y^0) > \frac{1}{4}$, then only two question are needed and the active set is empty at step three, etc.

Finally, since $C$, the code length, takes values in the non-negative integers $\{0, 1, \dots, \}$:

$$
\begin{aligned}
\tilde{C}_Q(X;Y) &:= \mathbb{E}[C] \\
&= \sum_{m=1}^{\infty} P(C \geq m) \\
&\leq \sum_{m=1}^{\infty} P(p(Y) < 2^{-(m-1)}) \\
&= \sum_{m=1}^{\infty} \sum_{y:p(y)<2^{-(m-1)}} p(y) \\
&= \sum_{y \in \mathcal{Y}} \sum_{m=1}^{\infty} 1_{\{p(y)<2^{-(m-1)}\}} p(k) \\
&= \sum_{y \in \mathcal{Y}} p(k)(1 - \log p(k)) \\
&= H(Y) + 1
\end{aligned}
$$

### A.7 Computing Mutual Information for (6)

#### A.7.1 Implementing $E^{IP}$: computing the first question $q_1$

Once the Decoder network has been learnt using variational inference, the first question $q_1$ can be calculated as per (6). Since the mutual information is completely determined by $p(q(x), y)$, which is obtained by numerically marginalizing the nuisances $Z$ from (8) using Monte Carlo integration. $\forall q \in Q$

$$
\begin{aligned}
p_{q(X)Y}(q(x), y) &= \int_\eta p_{Q(X)ZY}(Q(x), \eta, y) d\eta \\
&= \int_\eta p_{q(X)|ZY}(q(x) \mid \eta, y) p_Y(y) p_Z(\eta) d\eta \\
&\approx \frac{1}{N} \sum_{i=1}^{N} p_\omega(q(x) \mid y, \eta^{(i)}) p_Y(y)
\end{aligned}
\tag{20}
$$

In the last approximation, $p_\omega(q(x) \mid y, \eta^{(i)})$ is the distribution obtained using the trained Decoder network. $N$ is the number of i.i.d. samples drawn and $\eta^i \sim p_Z(\eta)$.

#### A.7.2 Derivation for (9)

.

$$
\begin{aligned}
p(q(x), y \mid B_{x,k}) &= \int p(q(x), \eta, y \mid B_{x,k}) d\eta \\
&= \int p(q(x) \mid \eta, y, B_{x,k}) p(\eta \mid y, B_{x,k}) p(y \mid B_{x,k}) d\eta. \\
&= \int p(q(x) \mid \eta, y) p(\eta \mid y, B_{x,k}) p(y \mid B_{x,k}) d\eta.
\end{aligned}
\tag{21}
$$

The first equality is an application of the law of total probability. The last equality appeals to the assumption that $\{q(X), q \in Q\}$ are conditionally independent given $Y, Z$ ((8)).

#### A.7.3 Recursive estimation of $p(y \mid B_{x,k})$

Finally, $p(y|B_k(x^0))$ (required for (9)) is computed recursively via the Bayes' theorem.

$$
\begin{aligned}
p(y \mid B_{x,k}) &\propto p(y, B_{x,k}) \\
&= p(q_k(x), y, B_{x,k-1}) \\
&\propto p(q_k(x) \mid y, B_{x,k-1}) p(y \mid B_{x,k-1})
\end{aligned}
\tag{22}
$$

$B_{x,0} = \emptyset$ (since no evidence via queries has been gathered from $x$ yet) and so $p(y \mid B_{x,0}) = p_Y(y)$. $p(y \mid B_{x,k})$ is obtained by normalizing the last equation in (22) such that $\sum_y p(y|B_{x,k}) = 1$. $p(q_k(x) \mid y, B_{x,k-1})$ can be estimated using (9).

## A.8 TRAINING THE CONDITIONAL INFERENCE NETWORK

A normalizing flow is a sequence of invertible transformations that takes a random variable from a simple source distribution (say Uniform or Gaussian) to an arbitrarily complex multi-modal target distribution. These invertible transformations are parameterized by deep neural networks that can express a richer family of distributions than the Gaussian/Uniform family. Specifically, we construct a sequence of invertible mappings of the form $\eta_l = \tilde{\Psi}_l(\{(q_i, q_i(x))\}_{1:k}, y, \eta_{l+1}; \gamma_l)$, each parameterized by a neural network with weights $\gamma_l$, such that $\eta_0$ is constrained to be normally distributed. The composite function $\eta_0 = \tilde{\Psi}(\{(q_i, q_i(x))\}_{1:k}, y, \eta; \gamma)$ is a composition of $N$ neural networks with parameters $\gamma = \{\gamma_l\}_{l \in \{1,2,...,N\}}$.

We call this network, $\tilde{\Psi}$, as the Conditional Inference Network. (Refer Fig. 2 for a pictorial depiction). By the change-of-variables formula for probability densities, $p(\eta \mid y, B_{x,k})$ can be written as.

$$
\begin{aligned}
p_\eta(\eta \mid y, B_{x,k}) = p_{\eta_0}(&\tilde{\Psi}(\{(q_i, q_i(x))\}_{1:k}, y, \eta; \gamma)) \\
&|\nabla_\eta \tilde{\Psi}(\{(q_i, q_i(x))\}_{1:k}, y, \eta; \gamma)|
\end{aligned}
\tag{23}
$$

To ensure $\tilde{\Psi}(\eta, y, \{(q_i, q_i(x))\}_{1:k}; \gamma)$ is invertible and the determinant in (23) is efficiently computable, the family of functions used is often constrained to those that admit an upper/lower triangular Jacobin. The normalizing flow model employed here is the realNVP model introduced in Dinh et al. (2016). For training we construct a dataset $\mathcal{D}^*$ from given dataset $\mathcal{D}$ (of $N$ i.i.d samples $\{x^i, y^i\} \sim p_{XY}(x, y)$) in the following manner.

1. Since we assumed $Q$ is finite, fix an enumeration. For every $(x, y)$ in $\mathcal{D}$, evaluate all the functions in $Q = \{q : q \in Q\}$ and obtain the sample $(Q(x), y)$. Sample $k \sim p_K(k)$, then sample $k$ random positions in $Q(x)$ as $m_k \sim p_M(m_k)$. $p_K(k)$ and $p_M(m_k)$ are user-defined distributions. In our applications $P_K(k)$ is taken to be $Poisson(\lambda = 10)$ and $p_M(m_k) = \mathcal{U}\{1, 2, ..., |Q|\}$

2. Sample $\eta \sim q_\phi(\eta|y, Q(x))$ using the trained Encoder network.

3. Mask $Q(x)$ according to positions in $m_k$ to obtain $k$-length sequence $\{(q, q(x))_{q \in \tilde{Q} \subseteq Q}\}$. In this way, we obtain samples from the desired joint $p(K, B_K(X), Y, \eta)$. $B_K(X)$ is a $K$-length random sequence $\{(q, q(X)) : q \in \tilde{Q}\}$ for some $\tilde{Q} \subseteq Q$.

The weights $\gamma$ are then learnt using stochastic approximation by optimizing the following objective using $\mathcal{D}^*$.

$$
\begin{aligned}
\max_\gamma \mathbb{E}_{K,B_K(X),Y,\eta}[\,&\log p_\eta(\eta|Y, B_K(X))] = \\
\max_\gamma \mathbb{E}_{K,B_K(X),Y,\eta}[\,&\log p_{\eta_0}(\tilde{\Psi}(B_K(X), Y, \eta; \gamma)) \\
&\log |\nabla_\eta \tilde{\Psi}(B_K(X), Y, \eta; \gamma)|]
\end{aligned}
\tag{24}
$$

The second equality is obtained by substituting (23) in (24). $p(\eta_0) = \mathcal{N}(0, I_d)$.

To understand the objective in 24, assume general random variables $\xi$ and $\psi$. Let our proposal distribution be $\tilde{p}(\psi|\xi)$ and the true distribution be $p(\psi|\xi)$. Consider a loss function

$$
\begin{aligned}
KL(p(\psi|\xi)||\tilde{p}(\psi|\xi)) = &\int p(\psi|\xi) log\, p(\psi|\xi) d\psi \\
&- \int p(\psi|\xi) log\, \tilde{p}(\psi|\xi) d\psi
\end{aligned}
\tag{25}
$$

Since (25) should hold $\forall \xi$, we take an expectation over $\xi$.

$$
\begin{aligned}
\mathbb{E}_\xi[KL(p(\psi|\xi)||\tilde{p}(\psi|\xi))] &= \int p(\xi)p(\psi|\xi)log\ p(\psi|\xi)d\psi d\xi \\
&\quad - \int p(\xi)p(\psi|\xi)log\ \tilde{p}(\psi|\xi)d\psi d\xi \\
&= \int p(\xi,\psi)log\ p(\psi|\xi)d\psi d\xi \\
&\quad - \int p(\xi,\psi)log\ \tilde{p}(\psi|\xi)d\psi d\xi
\end{aligned}
\tag{26}
$$

$$
\begin{aligned}
\min_{\tilde{p}(\psi|\xi)} &\ \mathbb{E}_\xi[KL(p(\psi|\xi)||\tilde{p}(\psi|\xi))] \\
&= \max_{\tilde{p}(\psi|\xi)} \int p(\xi,\psi)log\ \tilde{p}(\psi|\xi)d\psi d\xi \\
&= \max_{\tilde{p}(\psi|\xi)} \mathbb{E}_{\psi,\xi}[log\ \tilde{p}(\psi|\xi)]
\end{aligned}
\tag{27}
$$

The first term in (26) disappears since it does not depend on $\tilde{p}(\psi|\xi)$. Compare (27) and (24). Substitute $\psi := \eta$ and $\xi := (K, B_K(X), Y)$. The proposal $\tilde{p}(.)$ is parameterized by the Conditional Inference Network as $p_\eta(\eta|y, B_{x,k})$.

The issue is that we don't have access to the true distribution $p(\psi|\xi)$ to generate samples for a training set and learn an optimal $\tilde{p}(\psi|\xi)$ for each $\Xi = \xi$. However, we have access to the joint $p(\xi,\psi) := p(k, B_k(x), y, \eta)$ from which we could generate data $\mathcal{D}^*$ for optimizing (27) using stochastic approximation.

### A.9 NETWORK ARCHITECTURES AND TRAINING PROCEDURE

#### A.9.1 ENCODER AND DECODER NETWORKS

For the Information Pursuit Generative Model in subsection 3.2, we implemented a $\beta$-VAE as introduced in Higgins et al. (2017). The encoder-decoder architecture used is depicted in Fig. 4.

Notation for Fig. 4:

- L$(x, y)$ - Linear layer with $x$ input units and $y$ output units, followed by a BatchNorm layer and leakyRELU activation.
- C_$x$ - Convolution layer with $x$ $3 \times 3$ filters followed by a BatchNorm layer and leakyRELU activation.
- C_$x$_M - Same as C_$x$, followed by a Maxpool layer.
- DC_$x$_$y$_$z$ - Transposed Convolution layer with $x$ $y \times y$ filters and stride $z$, followed by a BatchNorm layer and leakyRELU activation.

Slope for leakyRELU activation was taken to be 0.3. BatchNorm and leakyRELU activation was not applied to the output layer of the encoder network (Fig. 4a). BatchNorm was not applied and the leakyRELU activation was replaced by Sigmoid in the output layer of the decoder network (Fig. 4).

**Training.** The $\beta$-VAE was trained by optimizing the Evidence Lower BOund (ELBO) objective,

$$
\begin{aligned}
\max_{\omega,\phi} \sum_{i=1}^n [&\mathbb{E}_{\eta \sim p'_\phi(\eta|y^{(i)},x^{(i)})}[\log p_\omega(x^{(i)}|\eta, y^{(i)})] \\
&- \beta KL(p'_\phi(\eta|y^{(i)}, x^{(i)})||p(\eta))].
\end{aligned}
\tag{28}
$$

using ADAM with learning rate 0.001 and momentum parameters $\beta_1 = 0.9$ & $\beta_2 = 0.999$. $\beta$ was taken as 4.0 for all experiments. The prior over latents $p_Z(\eta)$ is taken as $\mathcal{N}(0, I_{100})$ and $p_Y(y)$ estimated from the empirical distribution of the outputs from the training data. Data augmentation was performed on MNIST and its variants via elastic deformations. No Data augmentation was done for the other datasets. The network was trained for 200 epochs.

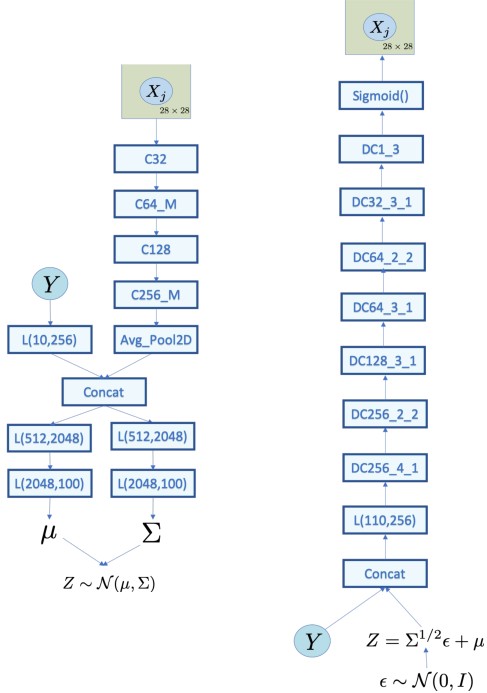

Figure 4: *The encoder-decoder architecture used in the $\beta$-VAE network.* Recall from section 3.2 4, $X_j$ are the image pixels, $Y$ is the class label and $Z$ are the nuisances. **(a) Encoder Network**: Takes the image $\{X_j\}_{j\in\{1,2,...,28\times28\}}$ and class label $Y$ as inputs and predicts the mean $\mu$ and diagonal covariance matrix $\Sigma$ of the nuisances $Z$. The weights of this network are denoted as $\omega$ in the main text; **(b) Decoder Network**: Takes the nuisance $Z$ and class label $Y$ as inputs and predicts the Bernoulli parameters of each pixel $X_j$ in the image. The weights of this network are denoted as $\phi$ in the main text. Best viewed in colour.

### A.9.2 CONDITIONAL INFERENCE NETWORKS

Unless stated otherwise, the notation for the figures in this section is the same as introduced in subsection A.9.1.

**Network Architecture.** For the Conditional Inference Network, introduced in Section 3.2, implemented a variant of the flow network introduced in Dinh et al. (2016). Fig. 5 depicts the overall architecture.

In Fig. 5 for our use-case with binary images and patch queries, $\{q_i, q_i(x)\}_{1:k}$, represents a masked image (sequence of patches observed), $y$ denotes the class label, $\eta$ are the nuisances distributed according to the target complex distribution and $\eta_0$ are the transformed random variables distributed according to a uni-modal standard Gaussian distribution. Each triplet of (Actnorm Layer, Permute Layer, Affine Layer) forms a layer of the flow network. The overall network is 25 layers deep. Each layer represents the function $\eta_l = \tilde{\Psi}_l(\{q_i, q_i(x)\}_{1:k}, y, \eta_{l+1}); \gamma_l)$, with weights $\gamma_l$. The weights of the flow network is denoted as $\gamma = \{\gamma_l\}_{l\in\{0,1,...,24\}}$. In what follows, we describe in detail each of the layers in a triplet.

*Actnorm Layer:* It has been proposed in literature to add a Batchnorm style layer with learnable scale $s$ and shift $b$ parameters to stabilize network optimization. Fig. 6 depicts the Actnorm layer network architecture used.

*Permute Layer:* This layer implements a permutation operation.

$$\eta_{l-1}^{permute} = P\eta_{l-1}^{actnorm} \tag{29}$$

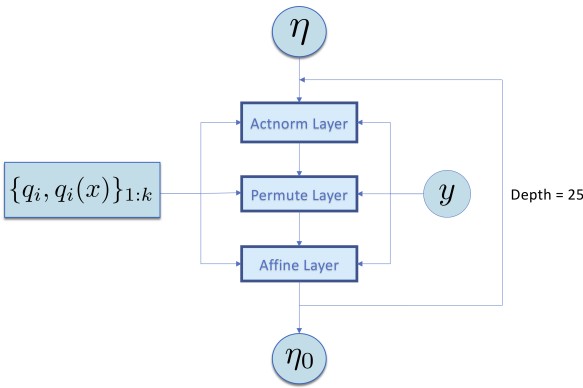

Figure 5: *Overall architecture of the Conditional Inference Network used based on RealNVP normalizing flows. The network takes $\{q_i, q_i(x)\}_{1:k}$, $Y$ and $\eta$ as inputs and performs 25 transformations $\eta \rightarrow \eta_{24} \rightarrow \eta_{23} \rightarrow \ldots \rightarrow \eta_0$. Each transformation is referred to as $\tilde{\Psi}_l := DNN(\{q_i, q_i(x)\}_{1:k}, y, \eta_{l+1}; \gamma_l)$ in Fig. 2.*

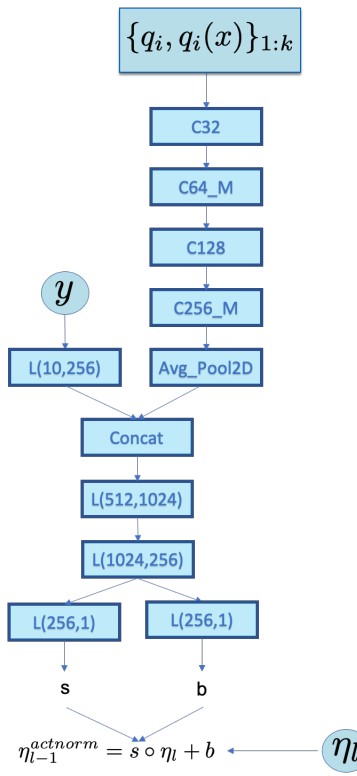

Figure 6: *The Actnorm layer used in each layer of the Conditional Inference Network. This layer takes $\eta_l, \{q_i, q_i(x)\}_{1:k})$ and $y$ as input and outputs $\eta_{l-1}^{actnorm}$. $\circ$ denotes element-wise multiplication.*

Here $P$ is a permutation matrix which swaps the first 50 dimensions of $\eta_{l-1}^{actnorm}$ with the next 50 dimensions. Recall, $\eta_l \in \mathbb{R}^{100} \ \forall l \in \{0, 1, \ldots, 25\}$.

***Affine Layer:*** This layer implements the following operations.

$$h_{l-1}^1, h_{l-1}^2 = \text{split}(\eta_{l-1}^{permute}) \tag{30}$$

$$s, t = NN_{\text{affine}}(h^1_{l-1}, B_k(x), y) \tag{31}$$

$$\begin{aligned} \alpha^1_{l-1} &= h^1_{l-1} \\ \alpha^2_{l-1} &= h^2_{l-1} \circ s + (1-s) \circ t \end{aligned} \tag{32}$$

$\circ$ denotes element-wise multiplication.

$$\eta_{l-1} = \text{concat}(\alpha^1_{l-1}, \alpha^2_{l-1}) \tag{33}$$

The split operation in (30) divides $\eta^{permute}_{l-1}$ into two halves, such that, $h^1_{l-1} = \eta^{permute}_{l-1}[0:50]$ and $h^2_{l-1} = \eta^{permute}_{l-1}[50:100]$.

Fig. 7 depicts the network architecture used for the $NN_{\text{affine}}$ in (31).

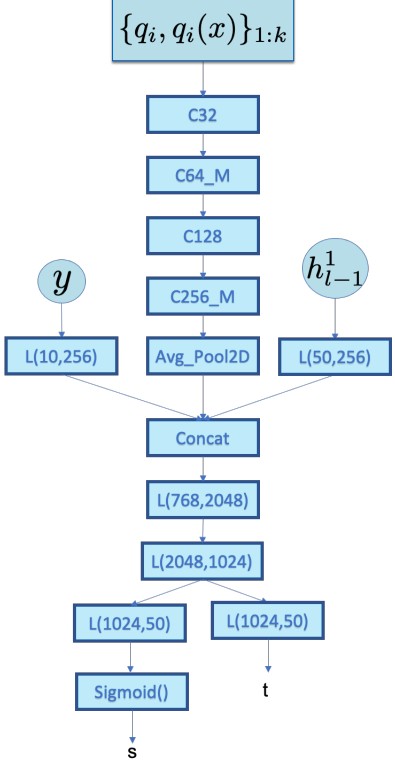

Figure 7: *The $NN_{affine}$ used in each Affine layer* (31) *of the Conditional Inference Network.* This layer takes $h^1_{l-1}, \{q_i, q_i(x)\}_{1:k}$ and $y$ as inputs and outputs scale vector $s$ and shift vector $t$.

Notice, each of the layers are invertible functions and hence their compositions (the 25 layer deep flow network) is also invertible, which is a key requirement for (23).

**Training.** The dataset for optimizing (24) was generated as outlined in section A.8 (denoted as $\mathcal{D}^*$). The Conditional Inference Network was trained by optimizing this objective using ADAM with learning rate 0.001 and momentum parameters $\beta_1 = 0.9$ & $\beta_2 = 0.999$. L2 regularization was added to stabilize the training and prevent gradients from exploding (a common problem in training normalizing flow networks). A scheduling was done for the regularization constant. Namely, we optimized (24) with L2 regularization parameter $\{10^{12}, 10^8, 10^2\}$ for 5 epochs each. Finally, the L2 regularization was relaxed (that is, regularization constant was made 0) and the network was trained for 500 epochs.

