# OpenReview forum: "Quantifying Task Complexity Through Generalized Information Measures"
_ICLR.cc/2021/Conference — Reject_

### Official Review · AnonReviewer2 · 2020-10-28

**Rating:** 5
**Confidence:** 3

**Review:**

The paper claims that existing measures of complexity such as entropy are not suitable for measuring task complexity since they focus on the complexity of X rather than the predictive relationship from X to Y. The paper argues that mutual information is not useful for comparing different learning tasks, as two tasks (MNIST vs Fashion-MNIST) can have similar MI but intuitively different complexity (second-to-last paragraph in related work). The paper proposes a measure for task complexity based on the number of queries required to predict a label of an input. The form of the queries is not specified, and the provided examples include half-space queries, single feature (decision-tree-like) queries, or high level semantic queries.  The proposed method instead considers a query generator E, then encoding X as the answers to the sequence of queries generated by E, and predicting Y from the answers. The complexity of a task is related to the number of equivalence classes induced by the input X and the query generator E.

I think the general premise is interesting, although I am not familiar with the related work (eg Achille, Tran) so I can't comment on how novel or different the idea of this paper is. Although the paper claims to have the first subjective notion of task complexity, to me this seems like more of a drawback since measures of complexity should be as standardized as possible, so as to allow comparison between different works. It may have been useful to cement down the three versions of complexity described on page 2, so future works can use them directly (for example provide details of measuring visual semantic complexity; and if it relies on extracting latent features of a neural net, i also wonder how close/different this is to [Achille]).

Also, the paper claims the drawback of Kolmogorov complexity is that it is not easily computable, but the methods described to compute the paper's proposed complexity are also highly involved, and requires multiple layers of approximations. It would have been helpful to have a more in-depth discussion of how (if at all) the proposed method is easier computationally compared to the other methods.

In proposition 1, epsilon seems not to have been motivated yet? The task complexity in Eq2 did not seem to have any notion of error, so why is there a probability of misclassification in prop 1?

The notation in Equation 1 looks off to me, since the RHS conditions on the set of all x's with the same encoding. Don't we want something to the effect of: p(y|x) = p(y|x')  for all x, x' such that A_q(x) = A_q(x') \forall q ?

The prefix-free constraint seems like it can be avoided if we just include q_STOP in the code? This seems more natural to me, and you can avoid the extra notation and sentences of explanation on the bottom of page 3.

Condition 2 in Prop3: we are assuming that y is categorical, and p_{Y|X=x} is a distribution over the labels in Y? Or does p_{Y|X=x} have all its probability mass on a single label? I assume it is the latter case, but writing it in terms of two inputs disagreeing \forall labels y is confusing when they are only assigned to one label each (in fact, shouldn't it be \exists y where x and x' disagree, instead of \forall y?).

I also didn't quite understand how the conditional inference network takes in {q, A_q(x)}_{1:k}. Isn't this not a fixed length sequence, in fact we don't even know the length of it beforehand since we decide when to stop at runtime based on the stopping condition? So in Fig2, how are the \Psi's able to take in variable-length sequences?

I'm curious about the exact cost of each iteration of the information pursuit algorithm. Given p(A_q(x), y | B) in Eq.9, do you compute p(A_q(x) | B) and p(y | B) exactly to get the mutual information, or do some sample-based approach? How many values can A_q(x) take on? And we need to do this for every possible query q \in Q in order to get argmax_q, so if A_q(x) can take on m values, are you doing O( |Q| m ) number of queries of the distribution p?

On the point of mutual information not being directly useful to predict difficulty in mapping X to Y, it seem that this paper "A Theory of Usable Information under Computational Constraints" Xu et al. [ICLR 2020] is very relevant. For example under a limited computational model, perhaps MNIST will have higher mutual information than Fashion-MNIST.

---

> ### Author Response · Authors · 2020-11-24
> **Response to AnonReviewer2-Part I**
>
> We thank the reviewer for their time and effort in providing valuable feedback on our paper and appreciate many insightful suggestions. We believe most of his/her comments are positive (or asking for clarification of technical details), and that we have adequately addressed the primary critique (standardization) raised in the review. We thus invite the reviewer to revise his/her rating of our work. More detailed responses follow.
>
> **I think the general premise is interesting ... i also wonder how close/different this is to [Achille]).**  \
> We are glad the reviewer found the ideas introduced in this paper interesting. However, we disagree that our definition based on a query set is a drawback because it does not lead to a standardized way of comparing tasks. First, given a query set Q that is suitable for multiple tasks, the complexity of all such tasks can be compared with respect to the same query set. Therefore, the query set actually standardizes the complexity measure. Second, we disagree that our work cannot be related to the work of others. For example, if Q is set of all possible binary functions of X, and Y is a function of X, then $C_Q(X; Y)$ is within 1 bit of the entropy of Y, $H(Y)$, which is precisely what Tran et al. define as complexity (hardness) of a task. However, as noted by the authors themselves, such a measure ignores dependencies between X and Y which is a crucial component of task complexity. By using different user-defined query sets, we make the proposed measure more task relevant. Third, we believe the work of Achille et al. is actually less standardized. In particular, Achille et al’s Task2Vec work uses the Fisher Information in the weights of an off-the-shelf neural network for measuring task complexity. Such a measure depends on the weights of the specific network architecture used (say, InceptionNet, or VGG). This would lead to different measures for the same task. Different off-the-shelf implementations of the same architecture could have wildly different pretrained weights. For example, Keras, pytorch and tensorflow implementations of pretrained InceptionNet have very different weights (https://arxiv.org/pdf/1801.01973.pdf).
> The example of visual semantic complexity does not rely on extracting latent features from a neural net  but rather on defining a query set that asks queries as in Visual Question Answering.
>
> **Also, the paper claims the drawback of Kolmogorov complexity ... compared to the other methods.**  \
> Kolmogorov Complexity is uncomputable whereas computing $C_Q(X; Y)$ is NP-Complete. The implication of this is that there exist dynamic programming based solutions that exactly compute $C_Q(X; Y)$. The complexity of these algorithms are typically exponential in $|Q|$ and so feasible only when $|Q|$ is small. For large $|Q|$, we must turn to approximations and Information Pursuit is one such strategy. In Section 3.1, we present a proposition citing sufficient conditions for certifying the quality of approximation for IP (upto estimation errors due to sampling). On the other hand, an algorithm for computing Kolmogorov Complexity does not exist, let alone an efficient one.
> Moreover, besides computability the more pressing issues with Kolmogorov complexity is that the measure is sensitive to dataset permutations which is clearly undesirable. Kolmogorov complexity also fails to distinguish between memorization and learning. A dataset sampled from $P_{XY}$ where Y is independent of X will have the maximum Kolmogorov based complexity measure. However, from a learning point of view there is nothing to learn - an optimal strategy is to simply predict $p(Y|X) = p(Y)$ regardless of the value of X! So, the task complexity of such tasks should be 0. Refer to https://arxiv.org/pdf/1904.03292.pdf for a more detailed discussion on this.
> The proposed measure $C_Q(X; Y)$ is not dataset permutation-sensitive since it is a property of the distribution $P_{XY}$. Secondly, $C_Q(X; Y) = 0$ when Y is independent of X (Proposition 2.1). We will add this discussion to the revised version of the paper.
>
> **In proposition 1 ... probability of misclassification in prop 1?** \
> We apologize for the confusion, this $\epsilon$ is different from the $\epsilon$ used in $\epsilon$-approximate task complexity. We will change it to $\delta$ in the revision. The main message of the proposition is to relate $C_Q(X; Y)$ with the number of equivalence classes of X induced by the optimal encoder. The probability of misclassification in prop 1 refers to the error of the upper bound we use to estimate the number of equivalence classes.
>
> **The notation in Equation 1 looks off to me, ... A_q(x) = A_q(x') \forall q ?** \
> Both these definitions are equivalent. We felt defining it this way is more intuitive since it says “the answers to all the queries $A_q(x) \ \forall q \in Q$ is sufficient to predict $Y$”. Informally this is the same as saying $p(y | x) = p(y | \\{A_q(x) \ \forall q \in Q\\})$.

---

> ### Author Response · Authors · 2020-11-24
> **Response to AnonReviewer2-Part II**
>
> **The prefix-free constraint seems ... on the bottom of page 3.** \
> We agree with the reviewer and will make this change in the revised version.
>
> **Condition 2 in Prop3: ... instead of \forall y?).** \
> We thank the reviewer for pointing this out and apologize for the typo. It should be $\exists y$ instead of $\forall y$. We have made this change in the revision.
>
> **I also didn't quite understand how the conditional inference network ... able to take in variable-length sequences?** \
> Depending on the query-set Q there are various ways of doing this. For our experiments, since Q was a set of 3x3 patches in binary images, we modelled $\\{q, A_q(x)\\}_{1:k}$ by masking out the parts of the image not contained in $\\{q, A_q(x)\\}_{1:k}$. In general, one could use a recurrent or attention-based architecture to compute a representation for  $\\{q, A_q(x)\\}_{1:k}$ which would be trained end-to-end along with $\Psi$. Training details of the conditional inference network is provided in Appendix A.8.
>
> **I'm curious about the exact cost ... the distribution p?** \
> Appendix A.7. provides details of computing $p(y|B)$ and $p(A_q(x)|B)$. $p(y|B)$ is computed recursively but still requires sampling to estimate $p(A_q(X) | y, B)$ (Refer Appendix A.7.3). $p(A_q(x) | B)$ is also estimated using sampling since the nuisances $\eta$ must be marginalized (refer the lines immediately following Equation 9).
> In general, if we estimate Mutual Information using $n$ samples and $A_q(x)$ can take $m$ values, then the cost of querying the distribution p is $O(|Q|nm)$. However, these computations are done in parallel on a GPU and so in principle, the complexity of sampling could be $O(1)$. In that case, the per iteration cost of the IP algorithm would be $O(|Q|)$ since one still needs to loop over all the queries to compute the argmax.
>
> **On the point of mutual information ... mutual information than Fashion-MNIST.** \
> We thank the reviewer for providing this citation. We would add this reference in the revised version. It might be interesting to see connections between $C_Q(X; Y)$ and $\mathcal{V}$-information in the future.

---

### Official Review · AnonReviewer3 · 2020-10-29
**An interesting paper**

**Rating:** 5
**Confidence:** 1

**Review:**

This proposes a new measurement for the complexity of learning tasks. The proposed method measures the complexity of a learning task by the minimum expected number of questions that need to be answered to solve the task.

Strengths:
The idea of using the minimum expected number of questions that need to be solved for measuring the task complexity is an interesting idea to me.
The paper provides theoretical justifications and connections with existing information theories.
The paper is generally clear and well constructed.

Weakness:
The experimental analysis is weak, only a simple case study is provided in the paper.

---

> ### Author Response · Authors · 2020-11-24
> **Response to AnonReviewer3**
>
> We are glad the reviewer found our work interesting. However, we do not believe the strength of the experimental analysis should be assessed based on quantity, but rather quality, level of difficulty of the experiments or whether it is an established problem. More specifically, extensive experiments might be needed to validate new ideas in well-established areas of research, especially in an area that addresses practical problems. On the other hand, purely theoretical papers may not need experiments at all, and few experiments might suffice for new areas that are not sufficiently well explored. We think that formal characterizations of task complexity are only beginning to emerge in recent years.

---

### Official Review · AnonReviewer4 · 2020-11-10
**Review of "Quantifying Task Complexity Through Generalized Information Measures"**

**Rating:** 6
**Confidence:** 5

**Review:**

This paper proposes a method to quantify the complexity of a learning task. The paper is motivated from the “20 questions“ game where an agent computes the answer (label) via a sequence of questions asked on the input data with answers given by an Oracle (simple functions of the data, in this case). The authors formalize this process and define the complexity of a learning task as the smallest number of questions from a given set Q necessary to predict the labels accurately averaged across the dataset. Information Pursuit (IP) of Geman & Jedynak 1996 is used to instantiate this definition using variational and normalizing-flow based models to learn the conditional distributions. Experimental results are shown for MNIST, Fashion-MNIST, KMNIST and Caltech Silhouettes datasets.

The main intellectual novelty of the paper is to define the complexity of a learning task using the number of questions. This comes with certain caveats that are discussed in the detailed comments below. While the paper is understandably a first step in this interesting program, more crisp experimental results are necessary before we can ascertain the utility of these ideas.

Detailed comments.

1. I have a philosophical gripe about this framework. It is widely observed that ensembles of decision trees which have been expanded until there is only sample at each leaf or, more recently, over-parameterized deep networks generalize better for machine learning tasks. The definition of task complexity developed in this paper does not relate to “learning” tasks. Indeed the complexity of the same task under decision made by above over-complete decision tree would be very large. The present paper is an attempt at computing the complexity of the conditional distribution p(Y | X) using a different “basis” that comes from the Oracle’s answers to the queries.
2. The complexity of a learning task should also be a function of the hypothesis class that is being used for the task. This is exactly the benefit for using quantities like VC-dimension. Why not, for instance, define the complexity of a task as the minimum-description-length (MDL) of the model that achieves at least a generalization gap of epsilon? Indeed, the prior over the hypothesis class in MDL is similar to the prior over the query set Q in this paper. There is recent work that captures the complexity of transfer learning while incorporating the hypothesis class, e.g., https://arxiv.org/abs/2011.00613, that the authors could seek synergies with.
3. The above two points are also seen in the claim about sub-additivity. Sub-additivity is a difficult property to have in general. If the tasks Y1 and Y2 conflict with each other, e.g., if they do not share any features and the model does not have sufficient capacity to learn both sets of features then the complexity of learning the two tasks simultaneously tasks should be _larger_ than the sum of their individual complexities.
4. The factorization in (8) need not be assumed. Since variational Bayes is used to approximate the true distribution on the left-hand side, one may simply say that the right-hand side is a particular variational family. This also applies to the paragraph above Fig. 3.
5. More refined experimental evidence is necessary before we can understand the merits of this definition. I find the current results difficult to appreciate, e.g., MNIST-0.05 and MNIST-0.1 should essentially have the same test error using any off-the-shelf CNN. Why is there is a gap between the relative test accuracy around, say, epsilon = 10 in Fig. 3a? I suspect this is an artifact of the variational/normalizing flow framework which does not learn good representations with noisy data and thereby results in a degradation of the validation error.
6. The learning task for classifying images in MNIST-translated should have the same complexity as that of MNIST because CNNs are translationally invariant. That this definition leads to a higher complexity for the former indicates that the setup where the agent searches for patches of input images is the real reason for this seeming increase in complexity.
7. It would be good to compare the ordering in complexity of these tasks using some other baseline method in the literature to compute the task distance, e.g., Task2Vec.

---

> ### Author Response · Authors · 2020-11-24
> **Response to AnonReviewer4-Part I**
>
> We thank the reviewer for their time and effort in providing valuable feedback on our paper and for providing a very detailed set of comments. However, at a high level we disagree with most of the reviewer’s critiques, which we believe arise from a disagreement between our definition of task complexity and the reviewer’s notion of complexity. We have provided very detailed responses to each one of the reviewer’s points, and we hope that our responses clarify the misunderstanding and that the reviewer revises his/her rating accordingly.
>
> Before responding to each point individually, let us clarify the misunderstanding by noting a key point about the notion of the complexity of a task.  Namely, a model (e.g., a linear classifier or a deep network) is *a particular strategy* for solving a task, but the complexity of solving a task with one particular choice of model is not the same as the complexity of the overall task itself.  Notions of model complexity, such as VC-dimension or Rademacher complexity are largely defined to explore how rich the model class is and produce performance guarantees for *a particular strategy* to solve a task.  This is at a fundamentally lower level than our goal here which is to define the complexity of the task as a whole, regardless of the choice of modeling strategy one makes in trying to solve the task.
>
> **Detailed Comments:** \
> **1. I have a philosophical gripe ... answers to the queries.** \
> To continue our discussion from above, we expand on several points. First, we disagree that our definition does not relate to “learning” tasks. On the contrary, we define learning a task as the problem of learning the conditional $p(Y | X)$, which we do by finding a minimal code $Code(X)$ for data $X$ such that $p(Y | X) = p(Y | Code(X))$, i.e., by construction the code is “sufficient” for learning the task. In fact, properties 2 and 3 in Proposition 2 allude to qualities expected from a complexity measure for learning tasks. Second, as stated above, our definition of task complexity is not based on any model class. It measures an intrinsic notion of complexity captured by the user-specified query set and is a property of the joint distribution $P(X,Y)$ (similar to how entropy is a property of the distribution). There is no generalization from finite samples involved. Thus the reviewer’s comment about over-complete decision trees having very large complexity (which is really speaking to the complexity of the model and not of the task) is not relevant to our notion of complexity.
>
> **2. The complexity of a learning task ... could seek synergies with.** \
> As a simple counter-example to the reviewer’s comment, the complexity of learning to dance is higher than the complexity of learning to walk and such a statement about the difference in complexity between two learning tasks is not a function of a measure of model complexity such as the VC dimension or MDL of the model. Therefore, we do not see dependence on any notion of model complexity (e.g., VC-dimension or MDL) as a requirement for defining task complexity, as the reviewer suggests. In this paper, we define learning a task as learning $p(Y | X)$. In this definition, we are not specifying a hypothesis class. In fact, our criticism of existing model complexity measures is precisely that they do not relate to the actual optimal predictor for the task. For example, the VC dimension of the hypothesis class of hyperplanes in R^d is d+1 regardless of whether we are using this hypothesis class to classify digits (MNIST) or natural objects (ImageNet). Also, there is no prior over the query set Q in our paper and so we do not understand the reviewer’s comment on it. Finally, we thank the reviewer for giving us the reference https://arxiv.org/abs/2011.00613. We will look into potential connections in future work.
>
> **3. The above two points are also seen ... sum of their individual complexities.** \
> As we argue above, the reviewer is thinking about task complexity in the context of the capacity of the model being used for learning the task, while we are defining a new notion of task complexity which does not have a specific model in mind. Going back to our earlier example, learning to dance is harder than learning to walk, and that statement has nothing to do with the capacity of the model used for learning. Alternatively, one can interpret our definition of task complexity as implicitly assuming that the model has sufficient capacity to learn the tasks. Therefore, the interpretation of the subadditivity property is that if $C_Q(X; Y_1, Y_2) = C_Q(X; Y_1) + C_Q(X; Y_2)$, then there is no shared structure between tasks $Y_1$ and $Y_2$. An implication of this would be exactly what the reviewer noted, models with lower capacity would be insufficient for jointly solving $Y_1$ and $Y_2$.

---

> ### Author Response · Authors · 2020-11-24
> **Response to AnonReviewer4-Part II**
>
> **4. The factorization in (8) ... the paragraph above Fig. 3.** \
> The factorization in (8) is not the variational approximation but the assumed true distribution of the observed and latent variables. The “forward” model in generative modelling. The variational distribution, which typically refers to the approximate posterior over latent nuisances, is $p'(\eta | y, A_Q(x))$.
>
> **5. More refined experimental evidence is necessary ... a degradation of the validation error.** \
> There appears to be misunderstanding regarding the point of this experiment. Off-the-shelf CNNs may have the same test error on MNIST-0.05 and MINIST 0.1, but that has nothing to do with what we are trying to show in Figure 3a. First of all, the x-axis in Figure 3a is not $\epsilon$, but rather $C^\epsilon_Q(X; Y)$, which is our approximate measure of complexity. Intuitively, $C^\epsilon_Q(X; Y)$ is the number of 3x3 patches that need to be observed to classify the 28x28 image with a small error. What Figure 3a shows is that as the amount of noise increases, more patches need to be observed to achieve the same classification error. For example, to achieve a relative error of 0.9, we need about 5 patches in MNIST, about 7 patches in MNIST-0.05, and about 8 patches in MNIST-0.1. Conversely, if we fix the number of patches to be only 5 and we try to predict the class only from those 5 patches, then the accuracy on clean patches has to be higher than the accuracy in noisier patches. We agree that if we allow the number of patches to be large enough so that the classifier gets to see the whole image ($C^\epsilon_Q(X; Y)$ = 80 patches), then the accuracy is the same for all tasks (as shown in the figure). Therefore, our experiments show precisely the merits of our proposed measure of complexity and that simple test error is not an appropriate measure of complexity. Finally, we do not think our experiments are an artifact of the variational/normalizing flow framework.
>
> **6. The learning task for classifying images ... seeming increase in complexity.** \
> As before, this comment pertains to a notion of task complexity which is not the one we propose. Specifically, the reviewer’s definition of task complexity here is based on test accuracy by a neural network, which we argue is not the correct notion of task complexity. In particular, while the argument that CNNs are translationally invariant is correct, there is prior knowledge about the task (translational invariance) whose complexity must also be accounted for. What if I didn’t know a priori that the task is translationally invariant? How would I discover that a convolutional architecture is needed? What is the extra complexity of discovering translational invariance? Again, we insist that our notion of complexity is really a complexity of the task itself (see dancing vs walking example) and not of the specific hypothesis class that is used to solve the task (e.g., a CNN).
>
> **7.It would be good to compare the ordering in complexity of these tasks using some other baseline method in the literature to compute the task distance, e.g., Task2Vec.** \
> The ordering of complexity for MNIST < Fashion-MNIST has been reported in prior work by Achille et. al. “The Information Complexity of Learning Tasks, their Structure and their Distance, 2019”. This ordering also correlates negatively with human performance on these two datasets, which could be considered as a proxy for task complexity independent of models. Specifically, human accuracy on MNIST is $\approx 0.998\%$ (https://arxiv.org/pdf/1202.2745.pdf) and on Fashion-MNIST is $0.835\%$ (https://arxiv.org/pdf/1202.2745.pdf).

---

### Author Response · Authors · 2020-11-24
**Summary of changes in rebuttal submission.**

In accordance with the reviews, we have fixed typos and made some minor modifications to the submission.

Some specific modifications:
1. To make the notation simpler we have changed $q$ to be the function in the query set $Q$ and $q(X)$ the answer evaluated at input $X$. The prior notation of $q$ being the index, $A_q$ being the function and $A_q(X)$ being the answer has been removed.

2. As suggested by *AnonReviewer2*; condition in Prop 2.3 has been corrected, a different symbol $\delta$ has been used in Prop 1, and removed the prefix-free constraint in the task complexity objective (Eq. 2).

3. To address a point raised by *AnonReviewer2* we have added a discussion about computability and drawbacks of Kolmogorov based complexity measures in the second last paragraph under Related Work.

4. We have added some examples to build intuition on the rationale behind our conditional independence assumptions in this work (Section 3.1 under "Information Pursuit Generative Model).

5. We have fixed an error in Figure 3a where the plot legend for MNIST-0.05 and MNIST-0.1 were erroneously swapped. The orange line corresponds to MNIST-0.05 and the green line corresponds to MNIST 0.1.

---

### Decision · Program_Chairs · 2021-01-07
**Final Decision**

**Decision:**

Reject

**Comment:**

This paper proposes a measure of task complexity based on a decision-DAG like "encoder" where we iteratively branch on some test on the input and the selection of future tests depends on the answer to previous tests until we reach a terminal node in the DAG.  We require that if $x$ and $x'$ reach the same terminal node then $P(y|x) = P(y|x')$.  The complexity of the task (the complexity of the distribution $p(x,y)$) is the minimum over all such DAGs of the expected depth of the terminal node for $x$ when drawing $x$ from the marginal $p(x)$.

The reviewers are not enthusiastic and I agree.